# CONDITIONAL INVERTIBLE NEURAL NETWORKS FOR GUIDED IMAGE GENERATION

## ABSTRACT

In this work, we address the task of natural image generation guided by a conditioning input. We introduce a new architecture called conditional invertible neural network (cINN). It combines the purely generative INN model with an unconstrained feed-forward network, which efficiently preprocesses the conditioning input into useful features. All parameters of a cINN are jointly optimized with a stable, maximum likelihood-based training procedure. Even though INNs and other normalizing flow models have received very little attention in the literature in contrast to GANs, we find that cINNs can achieve comparable quality, with some remarkable properties absent in cGANs, e.g. apparent immunity to mode collapse. We demonstrate these properties for the tasks of MNIST digit generation and image colorization. Furthermore, we take advantage of our bidirectional cINN architecture to explore and manipulate emergent properties of the latent space, such as changing the image style in an intuitive way. [1]

## 1 INTRODUCTION

Generative adversarial networks (GANs) produce ever larger and more realistic samples (Karras et al., 2017; Brock et al., 2019). Hence they have become the primary choice for a majority of image generation tasks. As such, their conditional variants (cGANs) would appear to be the natural tool for conditional image generation as well, and they have successfully been applied in many scenarios (Ledig et al., 2017; Miyato & Koyama, 2018). However, a lack in diversity is especially common when the condition itself is an image, and special precautions have to be taken to avoid mode collapse and training stability continues to pose a challenge.

Conditional variational autoencoders (cVAEs) do not suffer from the same problems. Training is generally stable, and since every data point is assigned a region in latent space, sampling yields the full variety of data seen during training. However cVAEs come with drawbacks of their own: The assumption of a Gaussian posterior on the decoder side implies an L2 reconstruction loss, which is known to cause blurriness. In addition, the partition of the latent space into diagonal Gaussians leads to either mode-mixing issues or regions of poor sample quality (Kingma et al., 2016). There has also been some success in combining aspects of both approaches for certain tasks, such as (Isola et al., 2017; Zhu et al., 2017b; Park et al., 2019).

We propose a third approach, by extending Invertible Neural Networks (INNs, Dinh et al.

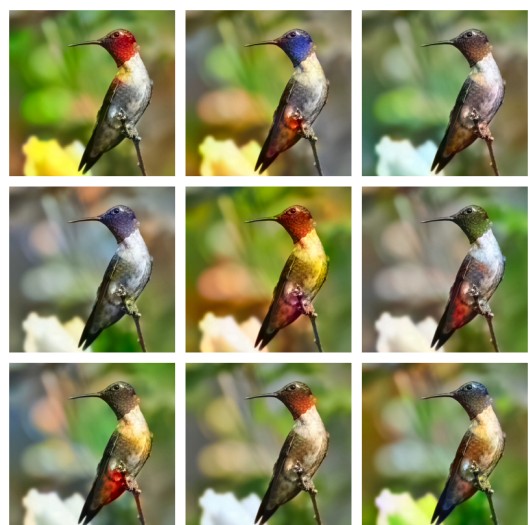

Figure 1: Diverse colorizations, which our network created for the same grayscale image. One of them shows ground truth colors, but which? Solution at the bottom of next page.

---

[1]Code is available, but held back for the review process due to anonymity.

(2016); Kingma & Dhariwal (2018); Ardizzone et al. (2019)) for the task of *conditional* image generation, by adding conditioning inputs to their core building blocks. INNs are neural networks which are by construction bijective, efficiently invertible, and have a tractable Jacobian determinant. They represent transport maps between the input distribution $p(\mathbf{x})$ and a prescribed, easy-to-sample-from latent distribution $p(\mathbf{z})$. During training, the likelihood of training samples from $p(\mathbf{x})$ is maximized in latent space, while at inference time, $\mathbf{z}$-samples can trivially be transformed back to the data domain. Previously, INNs have been used successfully for unconditional image generation, e.g. by Dinh et al. (2016) and Kingma & Dhariwal (2018).

Unconditional INN training is related to that of VAEs, but it compensates for some key disadvantages: Firstly, since reconstructions are perfect by design, no reconstruction loss is needed, and generated images do not become blurry. Secondly, each $\mathbf{x}$ maps to exactly one $\mathbf{z}$ in latent space, and there is no need for posteriors $p(\mathbf{z} \mid \mathbf{x})$. This avoids the VAE problem of disjoint or overlapping regions in latent space. In terms of training stability and sample diversity, INNs show the same strengths as autoencoder architectures, but with superior image quality. We find that these positive aspects apply to conditional INNs (cINNs) as well.

One limitation of INNs is that their design restricts the use of some standard components of neural networks, such as pooling and batch normalization layers. Our conditional architecture alleviates this problem, as the conditional inputs can be preprocessed by a *conditioning network* with a standard feed-forward architecture, which can be learned jointly with the cINN to greatly improve its generative capabilities. We demonstrate the quality of cINNs for conditional image generation and uncover emergent properties of the latent space, for the tasks of conditional MNIST generation and diverse colorization of ImageNet.

Given this, we believe that the cINN architecture brings the research field of INNs and other normalizing flow models a substantial step forward. Despite the fact that INNs have received very little attention in the literature in contrast to GANs, we find that cINNs can achieve comparable quality to cGANs, with some remarkable properties absent in cGANs. This includes diverse outputs by default, compared to sparse support or mode collapse in cGANs, easier explainability and interpretability of the learned latent representation, as well as simple and intuitive manipulation and interpolation of generated or existing images.

Our work makes the following contributions:

- We propose a new architecture called conditional invertible neural network (cINN), which combines an INN with an unconstrained feed-forward network for conditioning. It generates diverse images with high realism on par with existing approaces, while adding some noteworthy and useful properties.
- We demonstrate a stable, maximum likelihood-based training procedure for jointly optimizing the parameters of the INN and the conditioning network.
- We take advantage of our bidirectional cINN architecture to explore and manipulate emergent properties of the latent space. We illustrate this for MNIST digit generation and image colorization.

## 2 RELATED WORK

**Conditional Generative Modeling.** Modern generative models learn to transform noise (usually sampled from multivariate Gaussians) into desired target distributions. Methods differ by the model-family these transformations are picked from and by the losses determining optimal solutions.

Conditional generative adversarial networks (cGANs, Mirza & Osindero (2014)) train a pair of neural networks: a *generator* transforms a pair of conditioning and noise vectors to images, and a *discriminator* penalizes unrealistic looking images. The conditioning information is either concatenated to the noise (Mirza & Osindero, 2014), or fed into the network via conditional batch-norm layers (Dumoulin et al., 2017; Huang & Belongie, 2017; Park et al., 2019). Ensuring diversity of the generated images (for fixed conditioning) appears to be challenging in this approach. Recent BigGANs (Brock et al., 2019) successfully address this problem by using very large networks and batch sizes, but require parallel training on up to 512 TPUs. PacGANs (Lin et al., 2018) employ augmented discriminators, which evaluate entire batches of real or generated images together rather than one image at a time. CausalGANs (Kocaoglu et al., 2017) train two additional discriminator networks, called "labeler" and "anti-labeler", with the latter explicitly penalizing the lack of diversity.

Pix2pix (Isola et al., 2017) addresses the important special case when the target is conditioned on an image in a different modality, e.g. to generate satellite images from maps. In addition to the discriminator loss, it minimizes the L1 distance between generated and ground-truth targets using a paired training set, which contains corresponding images from both modalities. This leads to impressive image quality, but lack of diversity seems to be an especially hard problem in this case. In contrast, our method does not need explicit precautions to promote diversity.

Bidirectional architectures augment generator networks with complementary encoder networks that learn the generator's inverse and enable reconstruction losses, which exploit cycle consistency requirements. Conditional variational autoencoders (cVAEs, Sohn et al. (2015)) replace all distributions in a standard VAE (Kingma & Welling, 2013) by the appropriate conditional distributions, and are trained to minimize the evidence lower bound (ELBO loss). Since variational distributions are typically Gaussian, the reconstruction penalty is equivalent to squared loss, resulting in rather blurry generated images. This is avoided by AGE networks (Ulyanov et al., 2018) and CycleGANs (Zhu et al., 2017a), which combine standard cGAN discriminators with L1 reconstruction loss in the data domain, and bidirectional conditional GANs (Jaiswal et al., 2017), which extend the GAN discriminator to act on the distributions in data and latent space jointly. SPADE (Park et al., 2019), building upon pix2pix and pix2pixHD (Wang et al., 2018), augments cGANs with additional VAE encoders to shape the latent space such that diversity is ensured.

Instead of enforcing bijectivity through cycle losses, invertible neural networks are bidirectional by design, since encoder and generator are realized by forward and backward processing within a single bijective model. We focus on architectures whose forward and backward pass require the same computational effort. The coupling layer designs pioneered by NICE (Dinh et al., 2014) and RealNVP (Dinh et al., 2016) emerged as very powerful and flexible model families under this restriction. Using additive coupling layers, i-RevNets (Jacobsen et al., 2018b) demonstrated that the lack of information reduction from data space to latent space does not cause overfitting. The Glow architecture (Kingma & Dhariwal, 2018) combines affine coupling layers with invertible 1x1 convolutions and achieves impressive attribute manipulations (e.g. age, hair color) in generated faces images. This approach was recently generalized to video by Kumar et al. (2019).

Thanks to tractable Jacobian determinants, the coupling layer architecture enables maximum likelihood training (Dinh et al., 2014; 2016), but experimental comparisons with other training methods are inconclusive so far. For instance, Danihelka et al. (2017) found minimization of an adversarial loss to be superior to maximum likelihood training in RealNVPs, Schirrmeister et al. (2018) trained i-RevNets in the same manner as adversarial auto-encoders, i.e. with a discriminator acting in latent rather than data space, and Flow-GANs (Grover et al., 2018) performed best using bidirectional training, a combination of maximum likelihood and adversarial loss. On the other hand, maximum likelihood training worked well within Glow (Kingma & Dhariwal, 2018), and i-ResNets (Behrmann et al., 2018) could even be trained with approximated Jacobian determinants. In this work we reinforce the view that high-quality generative models can be trained by maximum likelihood loss alone. To the best of our knowledge, we are the first to apply the coupling layer design for *conditional* generative models, with the exception of Ardizzone et al. (2019), who use it to compute posteriors for (relatively small) inverse problems, but do not consider image generation. While Liu et al. (2019) recently proposed an effective post-hoc conditioning scheme for flow-based generative models, their method requires an additional cGAN to identify latent codes fulfilling the specified condition.

**Colorization.** State-of-the-art regression models for colorization produce visually near-perfect images (Iizuka et al., 2016), but do not account for the ambiguity inherent in this inverse problem. To address this, models would ideally define a conditional distribution of plausible color images for a given grayscale input, instead of just returning a single "best" solution.

Popular existing approaches for diverse colorization predict per-pixel color histograms from a U-Net (Zhang et al., 2016) or from hypercolumns of an adapted VGG network (Larsson et al., 2016). However, sampling from these local histograms independently can not lead to a spatially consistent colorization, requiring additional heuristic post-processing steps to avoid artefacts.

In terms of generative models, both VAEs (Deshpande et al., 2017) and cGANs (Isola et al., 2017; Cao et al., 2017) have been proposed for the task. However, their solutions do not reach the quality of the regression-based models, and cGANs in particular often lack diversity. To compensate, modifications and extensions to generative approaches have been developed, such as auto-regressive

models (Guadarrama et al., 2017) and CRFs (Royer et al., 2017). However, these methods are computationally very expensive and often unable to scale to realistic image sizes.

Conceptually closest to our proposed method is the work of (Ulyanov et al., 2018), where an encoder network maps color information to a latent space and a generator network learns the inverse transform, both conditioned on the grayscale image. Their experiments however are limited to a data set with only cars, and just three latent dimensions, leading to global, but no local diversity.

In contrast to the above, our flow-based cINN generates diverse colorizations in one standard feed-forward pass. It models the distribution of all pixels jointly, and allows for meaningful latent space manipulations.

## 3 METHOD

Our method is an extension of the affine coupling block architecture established by Dinh et al. (2016). There, each network block splits its input $\mathbf{u}$ into two parts $[\mathbf{u}_1, \mathbf{u}_2]$ and applies affine transformations between them that have strictly upper or lower triangular Jacobians:

$$\begin{aligned} \mathbf{v}_1 &= \mathbf{u}_1 \odot \exp\left(s_1(\mathbf{u}_2)\right) + t_1(\mathbf{u}_2) \\ \mathbf{v}_2 &= \mathbf{u}_2 \odot \exp\left(s_2(\mathbf{v}_1)\right) + t_2(\mathbf{v}_1) . \end{aligned} \qquad (1)$$

The outputs $[\mathbf{v}_1, \mathbf{v}_2]$ are concatenated again and passed to the next coupling block. The internal functions $s_j$ and $t_j$ can be represented by arbitrary neural networks, and are only ever evaluated in the forward direction, even when the coupling block is inverted:

$$\begin{aligned} \mathbf{u}_2 &= \left(\mathbf{v}_2 - t_2(\mathbf{v}_1)\right) \oslash \exp\left(s_2(\mathbf{v}_1)\right) \\ \mathbf{u}_1 &= \left(\mathbf{v}_1 - t_1(\mathbf{u}_2)\right) \oslash \exp\left(s_1(\mathbf{u}_2)\right) . \end{aligned} \qquad (2)$$

As shown by Dinh et al. (2016), the logarithm of the Jacobian determinant for such a coupling block is simply the sum of $s_1$ and $s_2$ over image dimensions.

### 3.1 CONDITIONAL INVERTIBLE TRANSFORMATIONS

We adapt the design of Eqs. (1) and (2) to produce a conditional version of the coupling block. Because the subnetworks $s_j$ and $t_j$ are never inverted, we can concatenate conditioning data $\mathbf{c}$ to their inputs without losing the invertibility, replacing $s_1(\mathbf{u}_2)$ with $s_1(\mathbf{u}_2, \mathbf{c})$ etc. Our conditional coupling block design is illustrated in Fig. 2.

In general, we will refer to a cINN with network parameters $\theta$ as $f(\mathbf{x}; \mathbf{c}, \theta)$, and the inverse as $g(\mathbf{z}; \mathbf{c}, \theta)$. For any fixed condition $\mathbf{c}$, the invertibility is given as

$$f^{-1}(\,\cdot\,; \mathbf{c}, \theta) = g(\,\cdot\,; \mathbf{c}, \theta). \qquad (3)$$

Figure 2: One conditional affine coupling block (CC).

### 3.2 MAXIMUM LIKELIHOOD TRAINING OF CINNS

By prescribing a probability distribution $p_Z(\mathbf{z})$ on latent space $Z$, the model $f$ assigns any input $\mathbf{x}$ a probability, dependent on both the network parameters $\theta$ and the conditioning $\mathbf{c}$, through the change-of-variables formula:

$$p_X(\mathbf{x}; \mathbf{c}, \theta) = p_Z\left(f(\mathbf{x}; \mathbf{c}, \theta)\right) \left| \det\left(\frac{\partial f(\mathbf{x}; \mathbf{c})}{\partial \mathbf{x}}\right) \right| . \qquad (4)$$

Here, we use the Jacobian matrix w.r.t. $\mathbf{x}$, $\partial f(\mathbf{x}; \mathbf{c})/\partial \mathbf{x}$. We will denote the determinant of the Jacobian, evaluated at some training sample $\mathbf{x}_i$, as $J_i \equiv \det(\partial f/\partial \mathbf{x}|_{\mathbf{x}_i})$. Bayes' theorem gives us the posterior over model parameters as $p(\theta; \mathbf{x}, \mathbf{c}) \propto p_X(\mathbf{x}; \mathbf{c}, \theta) \cdot p_\theta(\theta)$. Our goal is to find network parameters that maximize its logarithm, i.e. we minimize the loss

$$\mathcal{L} = \mathbb{E}_i\left[-\log\left(p_X(\mathbf{x}_i; \mathbf{c}_i, \theta)\right)\right] - \log\left(p_\theta(\theta)\right), \qquad (5)$$

which is the same as in classical Bayesian model fitting.

Inserting Eq. (4) with a standard normal distribution for $p_Z(\mathbf{z})$, as well as a Gaussian prior on the weights $\theta$ with $1/2\sigma_\theta^2 \equiv \tau$, we obtain

$$\mathcal{L} = \mathbb{E}_i\left[\frac{\|f(\mathbf{x}_i; \mathbf{c}_i, \theta)\|_2^2}{2} - \log|J_i|\right] + \tau\|\theta\|_2^2 . \qquad (6)$$

The latter term represents L2 weight regularization, while the former is the *maximum likelihood loss*. Training a network with this loss yields an estimate of the maximum likelihood network parameters $\hat{\theta}_{\text{ML}}$. From there, we can perform conditional generation for a fixed $\mathbf{c}$ by sampling $\mathbf{z}$ and using the inverted network $g$: $\mathbf{x}_{\text{gen}} = g(\mathbf{z}; \mathbf{c}, \hat{\theta}_{\text{ML}})$, with $\mathbf{z} \sim p_Z(\mathbf{z})$.

Training with the maximum likelihood method makes it virtually impossible for mode collapse to occur: If any mode in the training set has low probability under the current guess $p_X(\mathbf{x}; \mathbf{c}, \theta)$, the corresponding latent vectors will lie far outside the normal distribution $p_Z$ and receive big loss from the first L2-term in Eq. (6). In contrast, the discriminator of a GAN only supplies a weak signal, proportional to the mode's relative frequency in the training data, so that the generator is not penalized much for ignoring a mode completely.

### 3.3 CONDITIONING NETWORK

In complex settings, we expect that higher-level features of $\mathbf{c}$ need to be extracted for the conditioning to be effective, e.g. global semantic information from an image as in Section 4.2. In such cases, feeding the condition $\mathbf{c}$ directly into the cINN would place an unreasonable burden on the $s$ and $t$ networks, as higher-level features would have to be re-learned in each coupling block.

To address this issue, we introduce an additional feed-forward *conditioning network* $h$, which transforms the condition $\mathbf{c}$ to some intermediate representation $\tilde{\mathbf{c}} = h(\mathbf{c})$, and replace $\mathbf{c}_i$ in Eq. (6) with $\tilde{\mathbf{c}}_i = h(\mathbf{c}_i)$. The network $h$ can be pretrained, e.g. by using features from a VGG architecture trained for image classification. Alternatively or additionally, $h$ can be trained jointly with the cINN by propagating gradients from the maximum likelihood loss through the conditioning $\tilde{\mathbf{c}}$. In this case, the conditioning network will learn to extract features which are particularly useful for embedding the cINN inputs $\mathbf{x}$ into latent variables $\mathbf{z}$.

### 3.4 IMPORTANT DETAILS

For cINNs to match the performance of well-established architectures for conditional generation, we introduce a number of minor modifications and adjustments to the architecture and training procedure. With these adaptions, our training setup is very stable and converges every time.

**Noise as data augmentation.** We add a small amount of noise to the inputs $\mathbf{x}$ as part of the standard data augmentation. This helps to smooth out quantization artifacts in the input, and prevents sparse gradients when large parts of the image are completely flat (as e.g. in MNIST).

**Soft clamping of scale coefficients.** We apply an additional nonlinear function to the scale co-efficients $s$, of the form $s_{\text{clamp}} = (2\alpha/\pi)\arctan(s/\alpha)$, which yields $s_{\text{clamp}} \approx s$ for $|s| \ll \alpha$ and $s_{\text{clamp}} \approx \pm\alpha$ for $|s| \gg \alpha$. This prevents any instabilities stemming from exploding magnitude of the exponential $\exp(s_{\text{clamp}})$. We find $\alpha = 1.5$ work well. A similar scheme is used in the realNVP architecture (Dinh et al., 2016), whereby $\alpha$ is a learned vector, instead of a fixed hyperparameter. We find that a fixed $\alpha$ does not limit performance, while completely ruling out problems from exploding or vanishing scaling factors.

**Initialization.** Heuristically, we find that Xavier initialization (Glorot & Bengio, 2010) leads to stable training from the start. We experienced training instability when initial parameter values were too high. Similar to Kingma & Dhariwal (2018), we also initialize the last convolution in all $s$ and $t$ subnetworks to zero, so training starts from an identity transform.

**Soft channel permutations.** We use random orthogonal matrices to mix the information between the channels. This allows for more interaction between the two information streams $\mathbf{u}_1, \mathbf{u}_2$ in the coupling blocks. A similar technique was used by Kingma & Dhariwal (2018), but our matrices stay fixed throughout training and are guaranteed to be cheaply invertible.

**Haar wavelet downsampling.** All prior INN architectures use checkerboard patterns for reshaping to lower spatial resolutions. However, this possibly results in unnatural inputs to the following layers, where previously neighboring pixels are suddenly treated independently as distinct channels. We argue, supported by our ablation experiments, that a deliberately chosen spatial decomposition can be more natural, and more closely resemble the effect of pooling operations in feed-forward networks. For these reasons, we perform downsampling with Haar wavelets (Haar, 1910), which essentially

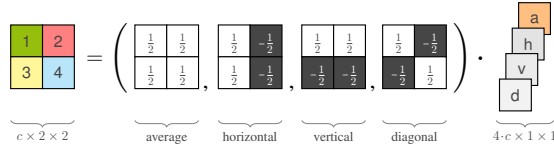

Figure 3: Haar wavelet downsampling reduces spatial dimensions & separates lower frequency content (a) from higher frequencies (h,v,d).

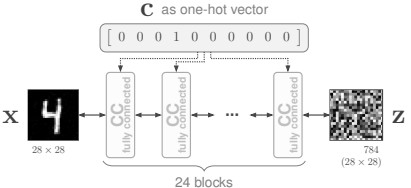

Figure 4: cINN model for conditional MNIST generation.

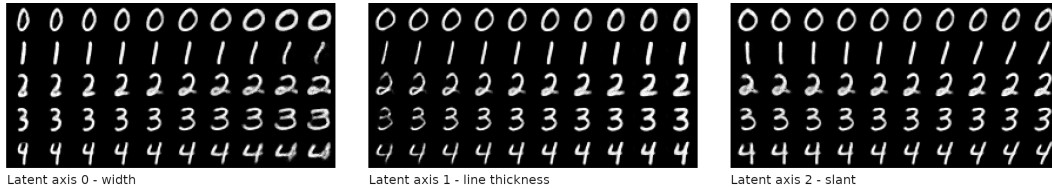

Latent axis 0 - width    Latent axis 1 - line thickness    Latent axis 2 - slant

Figure 5: Axes in our MNIST model's latent space, which linearly encode the style attributes width, thickness and slant.

decompose images into a $2 \times 2$ average pooling channel as well as vertical, horizontal and diagonal derivatives, see Fig. 3. This also motivates which channels should be split off to directly become latent variables, namely the derivative channels with high-resolution information, and which should be transformed further in later stages of the cINN, i.e. the pooling channels (see Sec. 4.2). The wavelet transform also contributes to mixing the variables between layers, complementing the soft permutations. Similarly, Jacobsen et al. (2018a) use a global discrete cosine transform as a final transformation in their INN, to replace global average pooling.

**Discussion.** The soft clamping, along with the Haar wavelet downsampling are critical for training stability at sufficiently high learning rates. With them, the network is exceptionally stable, and converges to a good optimum every time. The noise augmentation and initialization additionally increase the convergence speed. The permutations are necessary to achieve sufficient expressive power and high quality results, but do not affect the training stability or speed. Ablation curves for these improvements are given in the appendix.

## 4    EXPERIMENTS

We present results and explore the latent space of our models for two conditional image generation tasks: MNIST digit generation and image colorization.

### 4.1    CLASS-CONDITIONAL GENERATION FOR MNIST

As a first experiment, we perform simple class-conditional generation of MNIST digits. We construct a cINN of 24 coupling blocks using fully connected subnetworks $s$ and $t$, which receive the conditioning directly as a one-hot vector (Fig. 4). No conditioning network $h$ is used. For data augmentation we only add a small amount of noise to the images ($\sigma = 0.02$), as described in Section 3.4.

Samples generated by the model are shown in Fig. 6. We find that the cINN learns latent representations that are shared across conditions **c**. Keeping the latent vector **z** fixed while varying **c** produces different digits in the same style. This property, in conjunction with our network's invertibility, can directly be used for style transfer, as demonstrated in Fig. 7. This outcome is not obvious – the trained cINN could also decompose into 10 essentially separate subnetworks, one for each condition. In this case, the latent space of each class would be structured differently, and inter-class transfer of latent vectors would be meaningless. The structure of the latent space is further illustrated in Fig. 5, where we identify three latent axes with interpretable meanings. Note that while the latent space is learned without supervision, we found these axes in a semi-automatic fashion: We perform PCA on the latent vectors of the test set, without the noise augmentation. Since disentanglement is only unique up to a rotation in latent space, the directions corresponding to human-interpretable properties do not necessarily align with the latent axes. We therefore manually find a set of orthogonal vectors in the subspace of the first four PCA components which represent such properties.

Tidy

Slanted, narrow

Slanted left, wide

Messy

Faint

Bold

Figure 6: MNIST samples from our cINN conditioned on digit labels. All ten digits within one row $(0, \ldots, 9)$ were generated using the same latent code $\mathbf{z}$, but changing condition $\mathbf{c}$. We see that each $\mathbf{z}$ encodes a single style consistently across digits, while varying $\mathbf{z}$ between rows leads to strong differences in writing style.

Figure 7: To perform style transfer, we determine the latent code $\mathbf{z} = f(\mathbf{x}; \mathbf{c}, \theta)$ of a test image *(left)*, then use the inverse network $g = f^{-1}$ with different conditions $\hat{\mathbf{c}}$ to generate the other digits in the same style, $\hat{\mathbf{x}} = g(\mathbf{z}; \hat{\mathbf{c}}, \theta)$.

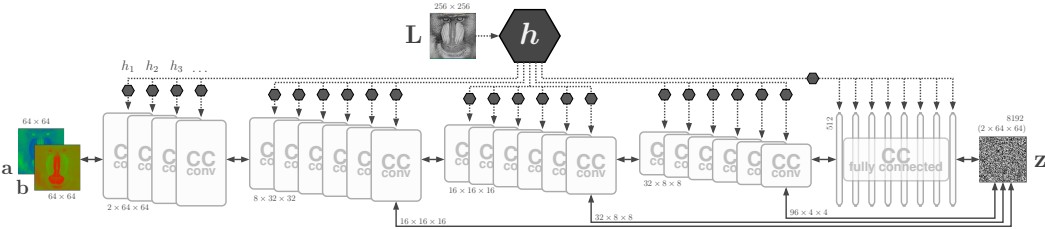

Figure 8: cINN model for diverse colorization. The conditioning network $h$ consists of a truncated VGG (Simonyan & Zisserman, 2014) pretrained to colorize ImageNet, with separate convolutional heads $h_1, h_2, h_3, \ldots$ tailoring the extracted features to each individual conditional coupling block (CC). After each group of coupling blocks, we apply Haar wavelet downsampling (Fig. 3) to reduce the spatial dimensions and, where indicated by arrows, split off parts of the latent code $\mathbf{z}$ early.

## 4.2 DIVERSE IMAGE COLORIZATION

For a more challenging task, we turn to colorization of natural images. The common approach for this task is to represent images in $Lab$ color space and generate color channels $\mathbf{a}, \mathbf{b}$ by a model conditioned on the luminance channel $\mathbf{L}$.

We train on the ImageNet dataset (Russakovsky et al., 2015), again adding low noise to the $\mathbf{a}, \mathbf{b}$ channels ($\sigma = 0.05$). As the color channels do not require as much resolution as the luminance channel, we condition on $256 \times 256$ pixel grayscale images, but generate $64 \times 64$ pixel color information. This is in accordance with the majority of existing colorization methods.

As with most generative INN architectures, we do not keep the resolution and channels fixed throughout the network, for the sake of computational cost. Instead, we use 4 resolution stages, as illustrated in Fig. 8. At each stage, the data is reshaped to a lower resolution and more channels, after which a fraction of the channels are split off as one part of the latent code. As the high resolution stages have a smaller receptive field and less expressive power, the corresponding parts of the latent vector encode local structures and noise. More global information is passed on to the lower resolution sections of the cINN.

For the conditioning network $h$, we start with the same VGG-like architecture from Zhang et al. (2016) and pretrain on the colorization task. By cutting off the network before the second-to-last convolution, we extract 256 feature maps of size $64 \times 64$ from the grayscale image $\mathbf{L}$. We then add independent heads on top of this for each conditional coupling block in the cINN, indicated by small hexagons in Fig. 8. Thus each coupling block $k$ receives its own specialized conditioning $\tilde{\mathbf{c}}_i^{(k)} = h_k\big(h(\mathbf{c}_i)\big)$. Each head consists of up to five strided convolutions, depending on its required output resolution, and a batch normalization layer. The ablation study in Fig. 14 confirms that the conditioning network is necessary to capture semantic information.

We initially train the cINN and the $h_k$, keeping the parameters of the conditioning network $h$ fixed, for 30 000 iterations. After this, we train both jointly until convergence, for 3 days on 3 Nvidia

| | cINN (ours) | | VAE-MDN | | cGAN | | PIC | | CNN | | BW | | Ground truth | |
|---|---|---|---|---|---|---|---|---|---|---|---|---|---|---|
| MSE (best of 8) | 3.53 | ±0.04 | 4.06 | ±0.04 | 9.75 | ±0.06 | 3.19* | ±0.09 | 6.77 | ±0.05 | – | | – | |
| LPIPS (best of 8) | 0.167 | ±0.001 | 0.171 | ±0.001 | 0.245 | ±0.001 | 0.147* | ±0.003 | 0.163 | ±0.001 | – | | – | |
| Variance (RGB) | 35.2 | ±0.3 | 21.1 | ±0.2 | 0.0 | ±0.0 | 44.1* | ± 1.3 | – | | – | | – | |
| Variance (LPIPS) | 0.1754 | ±0.001 | 0.0822 | ±0.001 | 0.0 | ±0.0 | 0.1755* | ±0.003 | – | | – | | – | |
| FID | 25.13 | ±0.30 | 25.98 | ±0.28 | 24.41 | ±0.27 | (Depends heavily on test set size) | | 24.95 | ±0.27 | 30.91 | ±0.27 | 14.69 | ±0.18 |
| VGG top 5 acc. | 85.00 | ±0.48 | 85.00 | ±0.48 | 84.62 | ±0.53 | 87.44* | ±1.2 | 86.86 | ±0.41 | 86.02 | ±0.43 | 91.66 | ±0.43 |

Table 1: Comparison of conditional generative models for diverse colorization (VAE-MDN: Desh-pande et al. (2017); cGAN: Isola et al. (2017)). We additionally compare to a state-of-the-art regression model ('CNN', no diversity, Iizuka et al. (2016)), and the grayscale images alone ('BW'). For each of 5k ImageNet validation images, we compare the best pixel-wise MSE of 8 generated colorization samples, the pixel-wise variance between the 8 samples as an approximation of the diversity, the Fréchet Inception Distance (Heusel et al., 2017) as a measure of realism, and the top 5 accuracy of ImageNet classification performed on the colorized images, to check if semantic content is preserved by the colorization. * Note: It was not possible to process all test images with PIC, due to the long inference times. Only 768 images are used. Full results will be included in the camera ready version.

GTX1080 GPUs. The Adam optimizer is essential for fast convergence, and we lower the learning rate when the maximum likelihood loss levels off.

At inference time, we use joint bilateral upsampling (Kopf et al., 2007) to match the resolution of the generated color channels $\hat{a}, \hat{b}$ to that of the luminance channel $L$. This produces visually slightly more pleasing edges than bicubic upsampling, but has little to no impact on the results. It was not used in the quantitative results table, to ensure an unbiased comparison.

Latent space interpolations and color transfer are shown in Figs. 9 and 10, with more experiments in the appendix. In Table 1, a quantitative comparison to existing methods is given. While on par with VAE-MDN in other metrics, the cINN clearly has the best sample diversity, as summarized by the variance and best-of-8 accuracy. The cGAN completely ignores the latent code and relies only on the condition – there is no diversity, in line with results from Isola et al. (2017). In terms of FID score, the cGAN performs best, although its results do not appear more realistic to the human eye, cf. Fig. 13. This may be due to the fact that FID is sensitive to outliers, which are unavoidable for a truly diverse method (see Fig. 12), or because the discriminator loss implicitly optimizes for the similarity of deep CNN activations. The VGG classification accuracy of colorized images is decreased for all generative methods equally, because occasional outliers may lead to misclassification.

Because the diverse COLORGAN by Cao et al. (2017) was exclusively trained on the LSUN bedrooms dataset (Yu et al., 2015) at small resolution, and would not converge on ImageNet, we compare this separately and provide results in the appendix. In this simple case, the performance of the cINN and COLORGAN is almost equal in all apsects, with much shorter training time for the cINN.

## 5 CONCLUSION AND OUTLOOK

We have proposed a conditional invertible neural network architecture which enables guided generation of diverse images with high realism. For image colorization, we believe that even better results can be achieved when employing the latest tricks from large-scale GAN frameworks. Especially the non-invertible nature of the conditioning network makes cINNs a suitable method for other computer vision tasks such as diverse semantic segmentation.

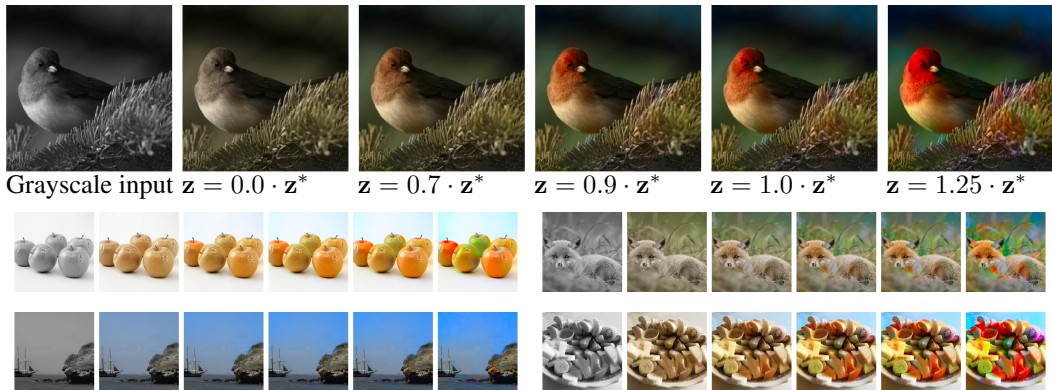

Figure 9: Effects of linearly scaling the latent code $\mathbf{z}$ while keeping the condition fixed. Vector $\mathbf{z}^*$ is "typical" in the sense that $\|\mathbf{z}^*\|^2 = \mathbb{E}\big[\|\mathbf{z}\|^2\big]$, and results in natural colors. As we move closer to the center of the latent space ($\|\mathbf{z}\| < \|\mathbf{z}^*\|$), regions with ambiguous colors become desaturated, while less ambiguous regions (e.g. sky, vegetation) revert to their prototypical colors. In the opposite direction ($\|\mathbf{z}\| > \|\mathbf{z}^*\|$), colors are enhanced to the point of oversaturation.

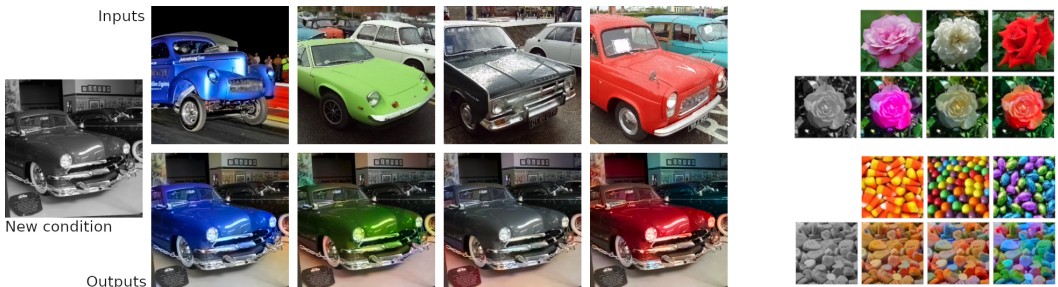

Figure 10: For color transfer, we first compute the latent vectors $\mathbf{z}$ for different color images $(\mathbf{L}, \mathbf{a}, \mathbf{b})$ *(top row)*. We then send the same $\mathbf{z}$ vectors through the inverse network with a new grayscale condition $\mathbf{L}^*$ *(far left)* to produce transferred colorizations $\mathbf{a}^*, \mathbf{b}^*$ *(bottom row)*. Differences between reference and output color (e.g. pink rose) can arise from mismatches between the reference colors $\mathbf{a}, \mathbf{b}$ and the intensity prescribed by the new condition $\mathbf{L}^*$.

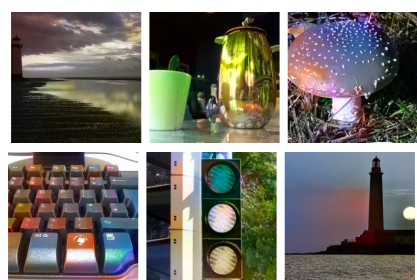

Figure 12: Failure cases of our method. *Top:* Sampling outliers. *Bottom:* cINN did not recognize an object's semantic class or connectivity.

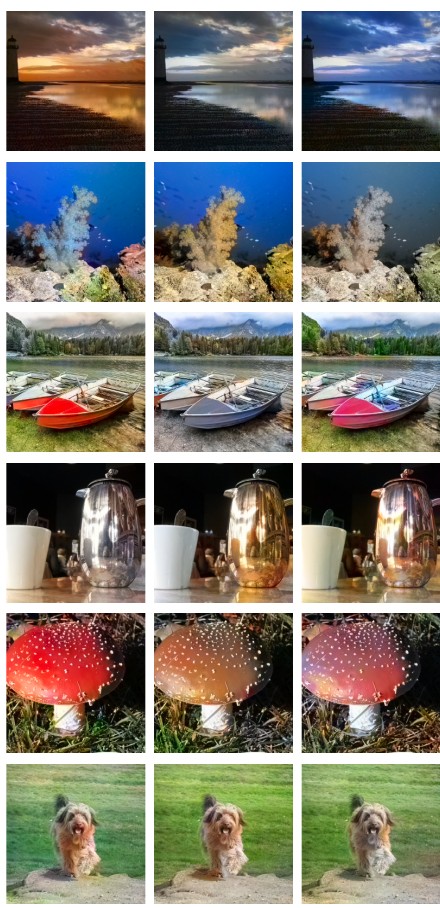

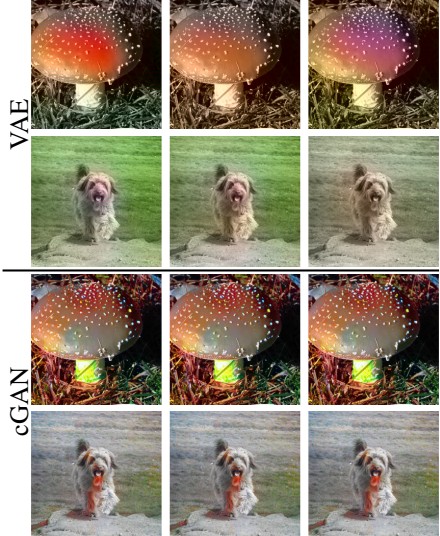

Figure 11: Diverse colorizations produced by our cINN.

Figure 13: Alternative methods have lower diversity or quality, and suffer from inconsistencies within objects, or color blurriness and bleeding (compare Fig. 11, bottom).

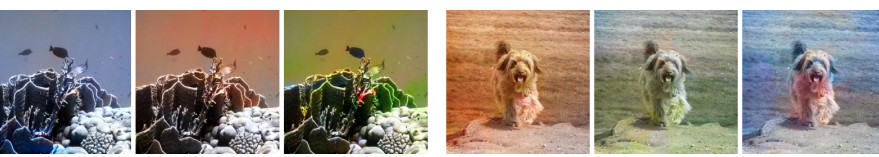

Figure 14: In an ablation study, we train a cINN using the grayscale image directly as conditional input, without a conditioning network $h$. The resulting colorizations largely ignore semantic content which leads to exaggerated diversity. More ablations are found in the appendix.

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

# A ADDITIONAL EXPERIMENTS

## A.1 ABLATION OF TRAINING IMPROVEMENTS

To demonstrate the improved stability and training speed through the improvements from Sec. 3.4, we perform ablations, see Fig. 15. The ablations for colorization were performed for the LSUN bedrooms task, due to training speed.

We find that for stable training at Adam learning rates of $10^{-3}$, the clamping and Haar wavelet downsampling are strictly necessary. Without these, the network has to be trained with much lower learning rates and more careful and specialized initialization, as used e.g. in Kingma & Dhariwal (2018). Beyond this, the noise augmentation and permutations lead to the largest improvement in final result. The effect of the noise is more pronounced for MNIST, as large parts of the image are completely black otherwise. For natural images, dequantization of the data is likely to be the main advantage of the added noise. Note however, that the training curves of the models with and without noise augmentation is not directly comparable, as the loss differs an additional summand $\approx \log(\sigma_{\text{aug.}})$. The effect on the training speed and stability is clearly visible regardless. The initialization only improves the final result by a small margin, but also converges noticeably faster.

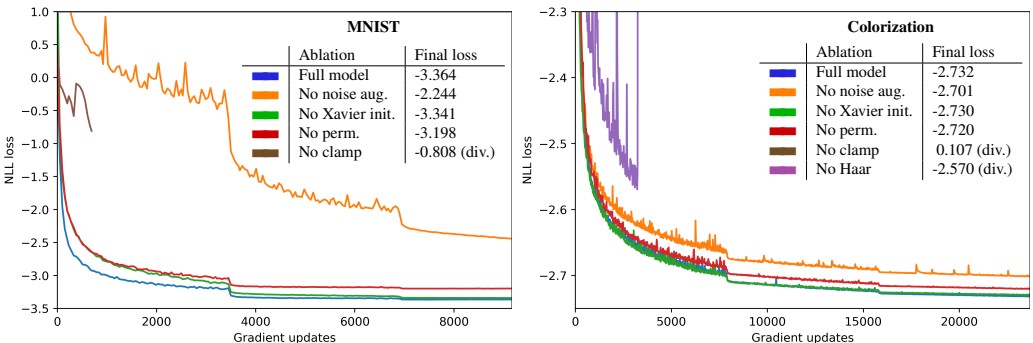

Figure 15: Training curves for each task, ablating the different improvements. "div." denotes that the training diverges, and the lowest loss so far is given.

## A.2 LSUN BEDROOMS

To provide a simpler model for more in-depth experiments and ablations, we additionally train a cINN for colorization on the LSUN bedrooms dataset Yu et al. (2015). We use a smaller model than for ImageNet, and train the conditioning network jointly from scratch, without pretraining. Both the conditioning input, as well as the generated color channels have a resolution of $64 \times 64$ pixels. The entire model trains in under 4 hours on a single GTX 1080Ti GPU.

To our knowledge, the only diversity-enforcing cGAN architecture previously used for colorization is the colorGAN Cao et al. (2017), which is also trained exclusively on the bedrooms dataset. Training the colorGAN for comparison, we find it requires over 24 hours to converge stably, after multiple restarts. The results are generally worse than those of the cINN, as shown in Fig. 16. While the resulting pixel-wise color variance is slightly higher for the colorGAN, it is not clear whether this captures the true variance, or whether it is due to unrealistically colorful outputs, such as in the second row in Fig. 16.

| Metric | cINN | colorGAN |
|---|---|---|
| MSE best-of-8 | **6.14** | 6.43 |
| Variance | 33.69 | **39.46** |
| FID | **26.48** | 28.31 |

Table 2: Quantitative comparison between smaller cINN and colorGAN on LSUN bedrooms. The metrics used are explained in Table 1.

cINN                                    COLORGAN

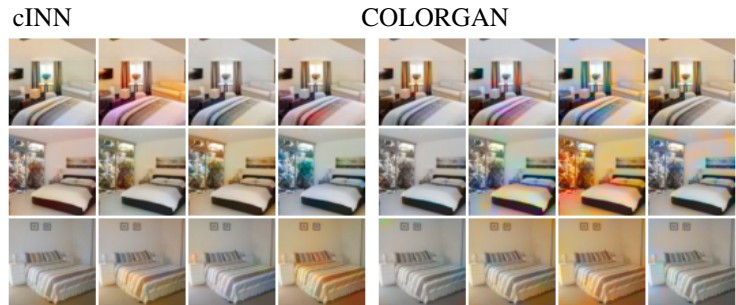

Figure 16: Qualitative comparison between smaller cINN and colorGAN on LSUN bedrooms.

## B  ADDITIONAL FIGURES

### B.1  COLORIZATION – TRANSFER

The following figure contains an additional example of the color transfer, shown in Fig. 9 in the paper.

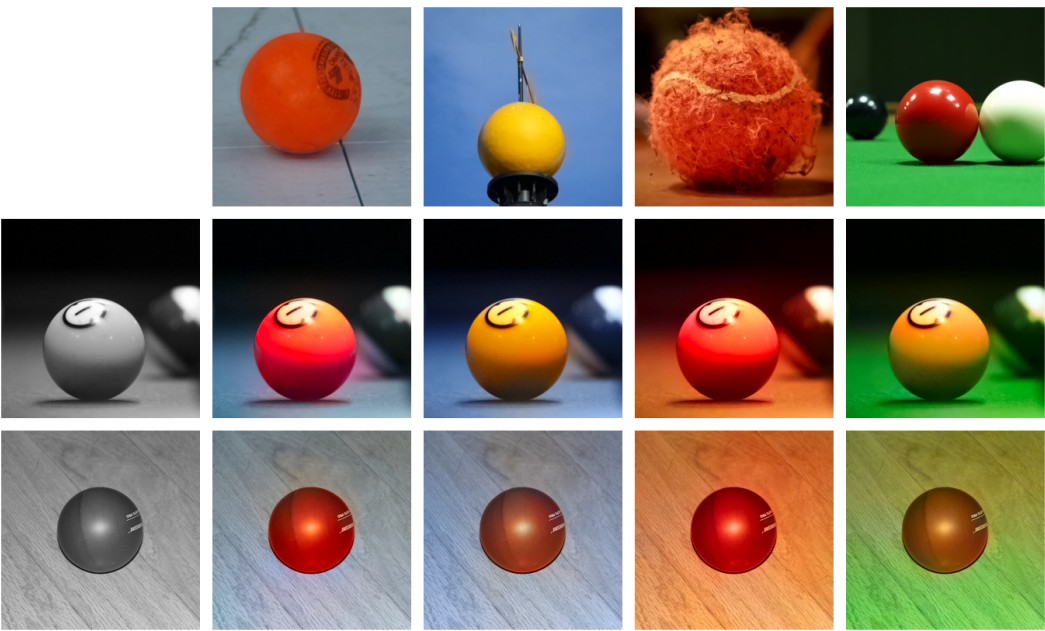

### B.2  COLORIZATION – INTERPOLATIONS

In the following, we show 2-dimensional interpolations in latent space. Two random latent vectors $\mathbf{z}^{(1)}$, $\mathbf{z}^{(2)}$ are linearly combined:

$$\mathbf{z}^* = a_1 \mathbf{z}^{(1)} + a_2 \mathbf{z}^{(2)}$$

with varying $a_1$, $a_2 \in [-0.9 \dots 0.9]$ across each axis of a grid. The center image has $\mathbf{z}^* = 0$. Note that the images in the corners have a larger magnitude than trained for, $\|z^*\|_2 \approx 1.3\,\mathbb{E}\big[\|\mathbf{z}\|_2\big]$, leading to some oversaturation artifacts, as in Fig. 12 of the main paper.

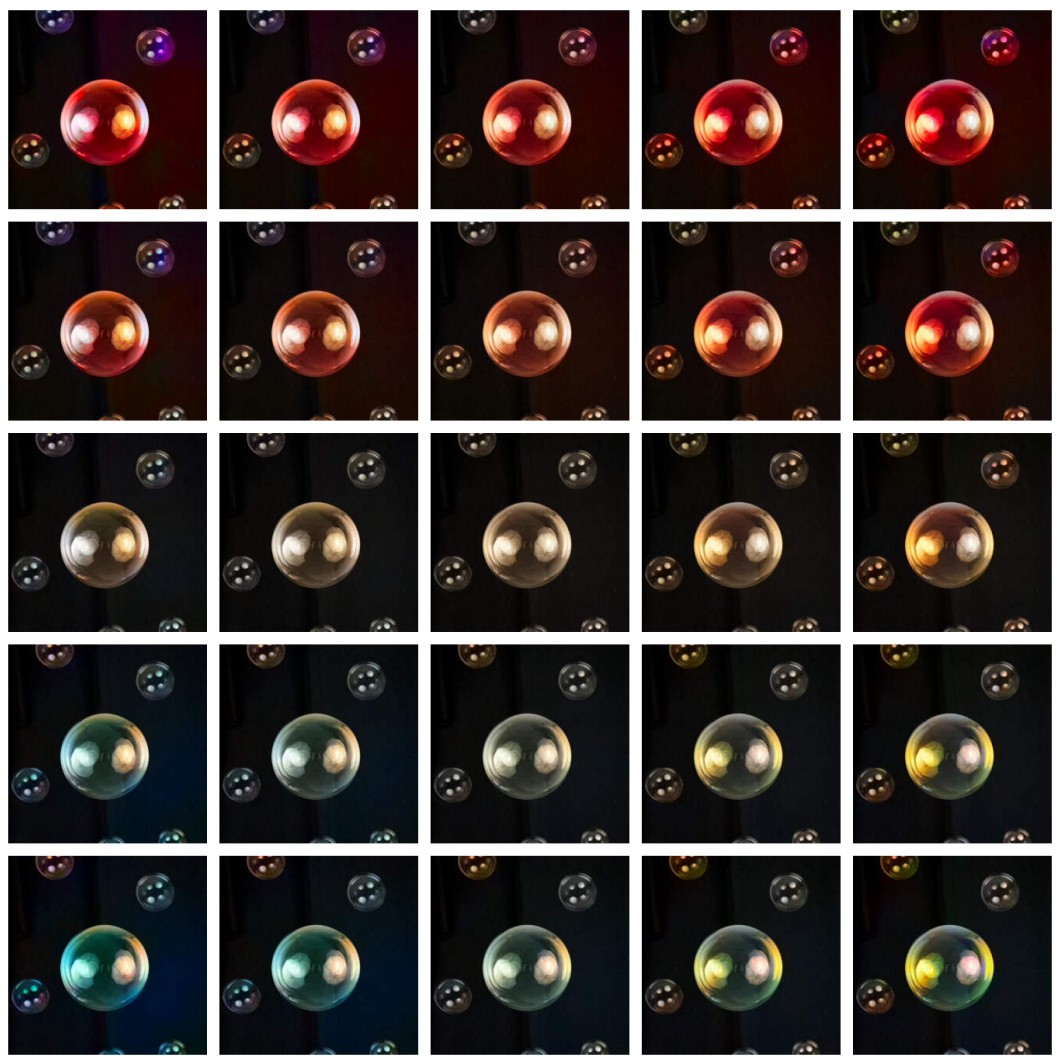

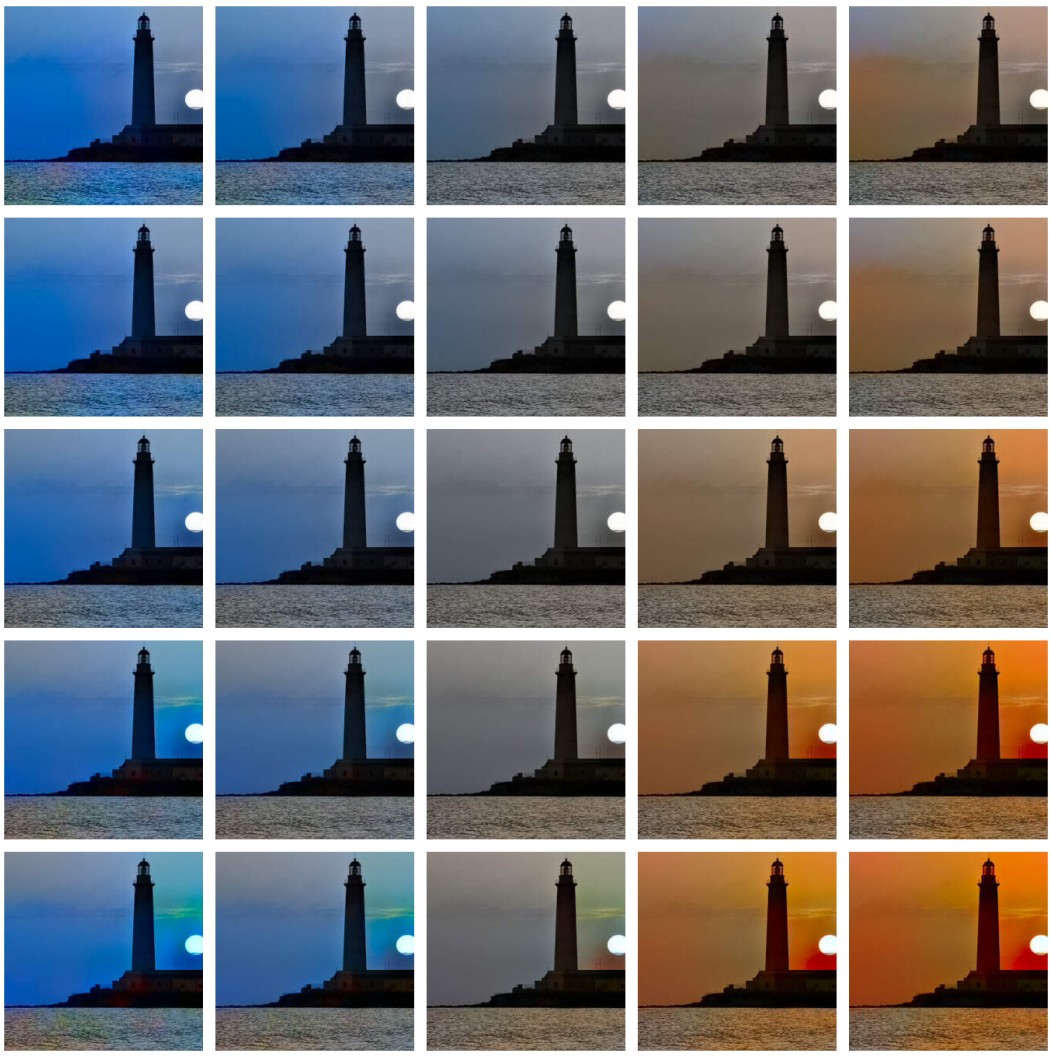

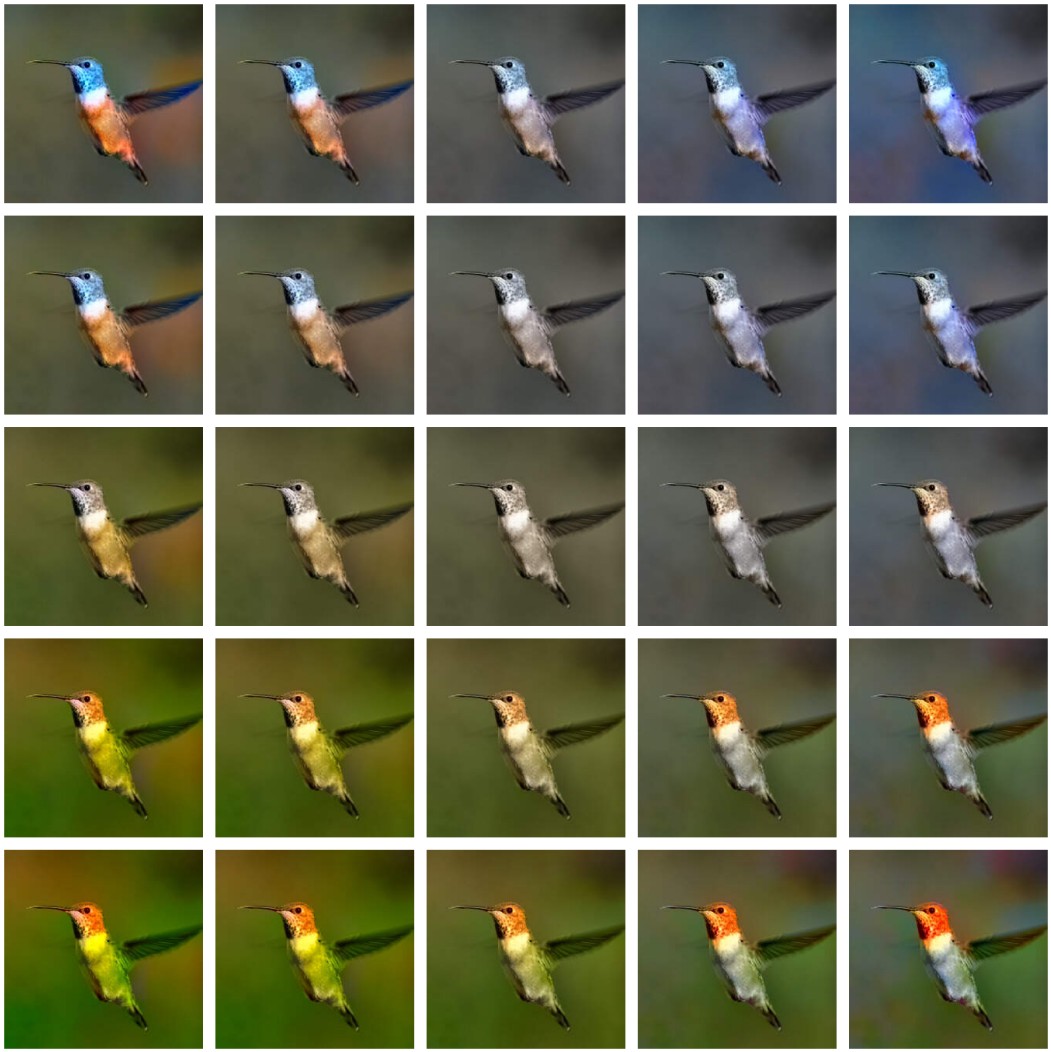

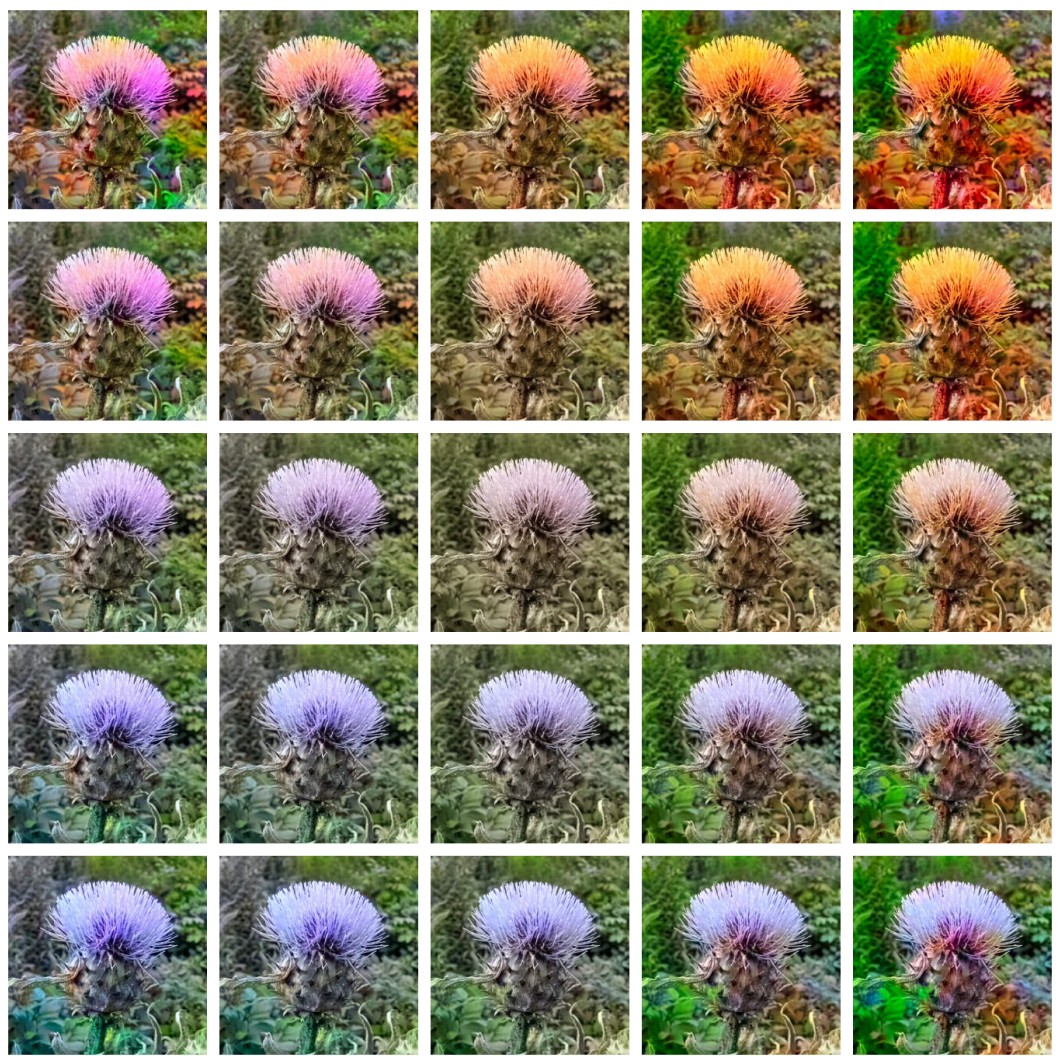

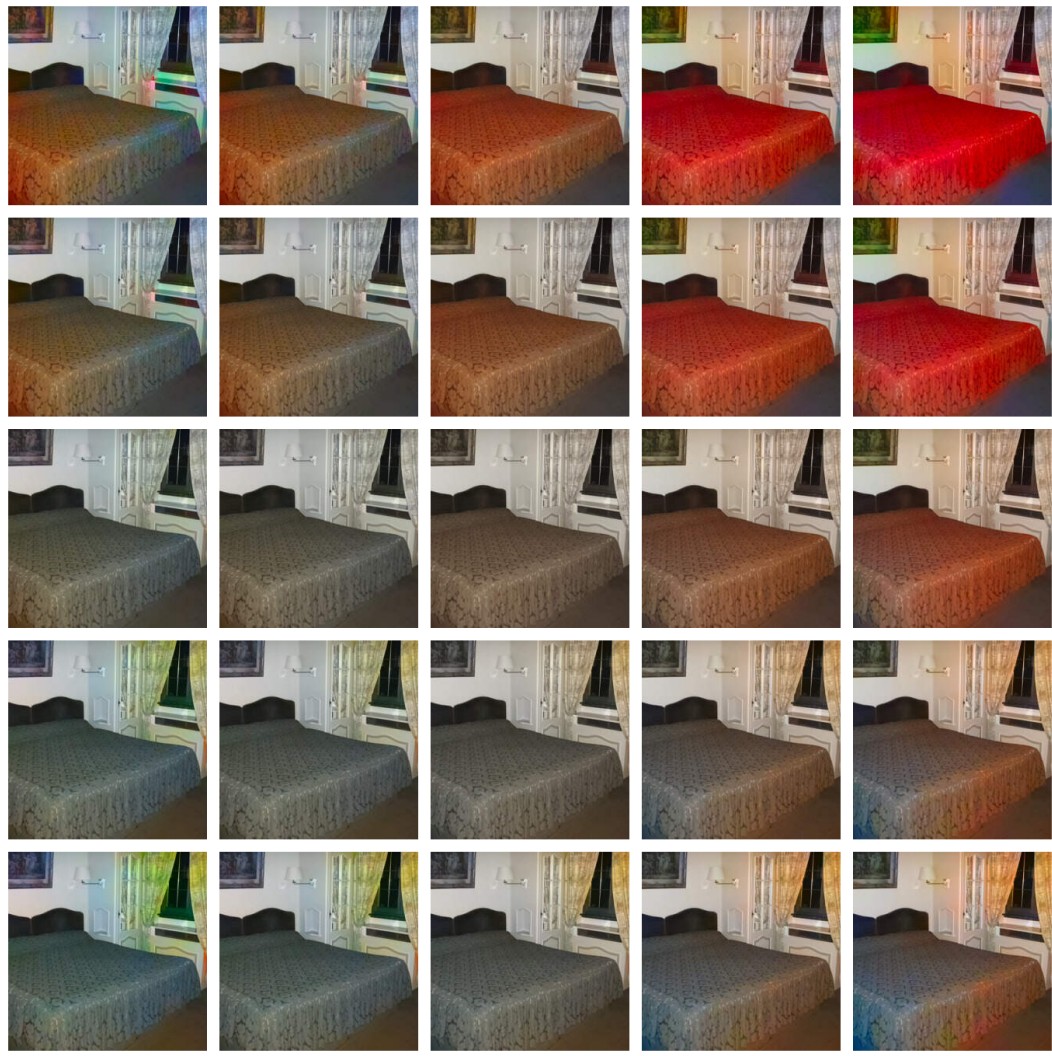

## B.3 Colorization – Additional examples

On the following pages, we provide some additional colorized images, as well as comparisons to alternative methods. All images are taken from the ImageNet 2012 validation set, and all methods were trained on ImageNet 2012. As we do not observe any significant diversity for the cGAN, we only provide a single sample.

### B.3.1 General examples

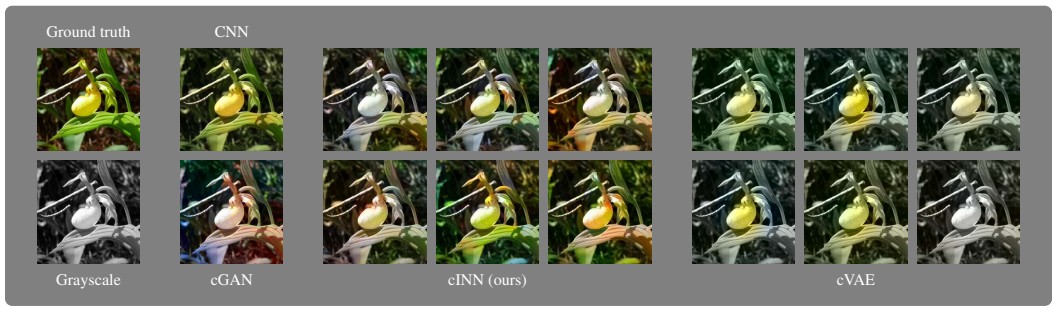

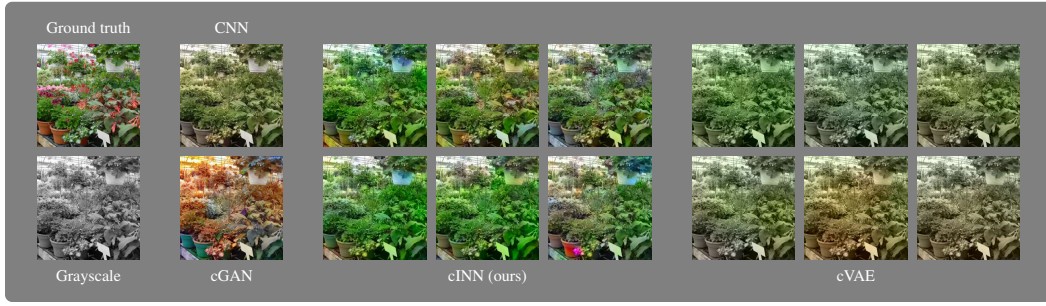

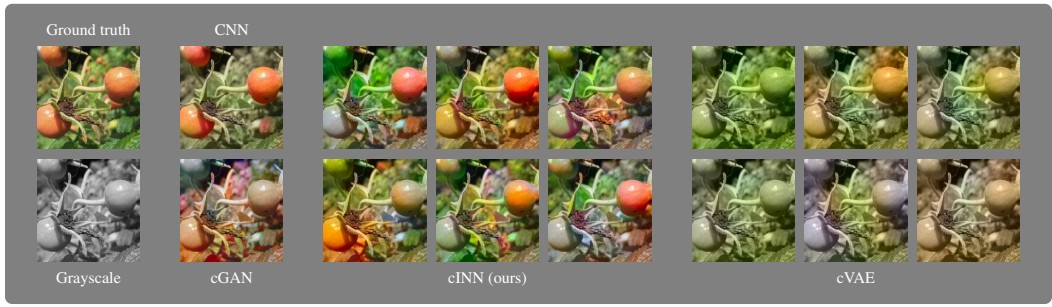

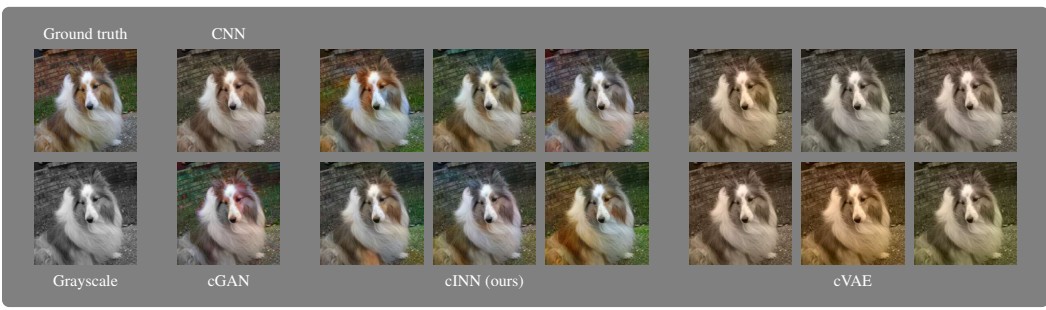

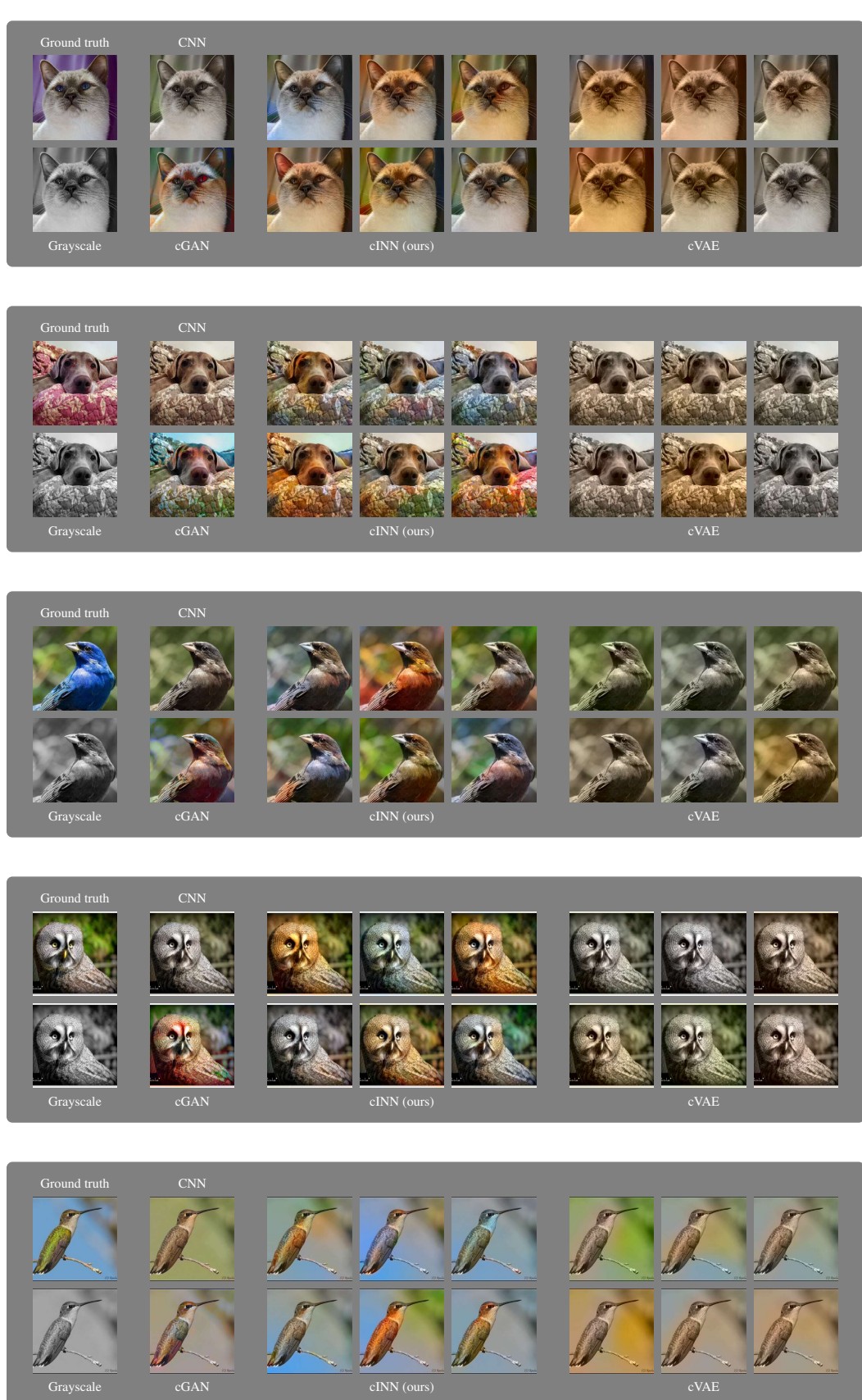

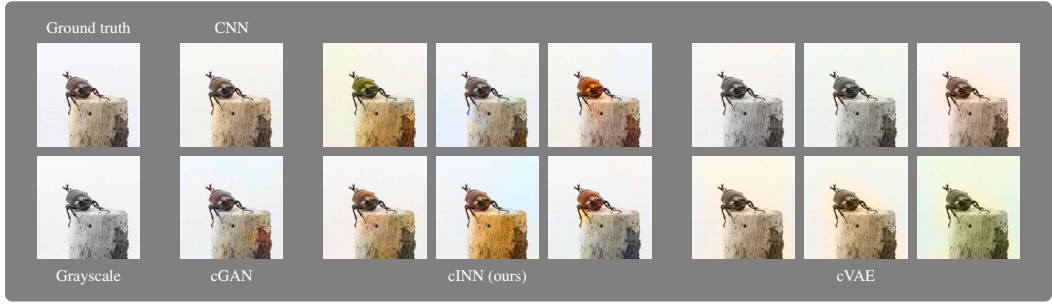

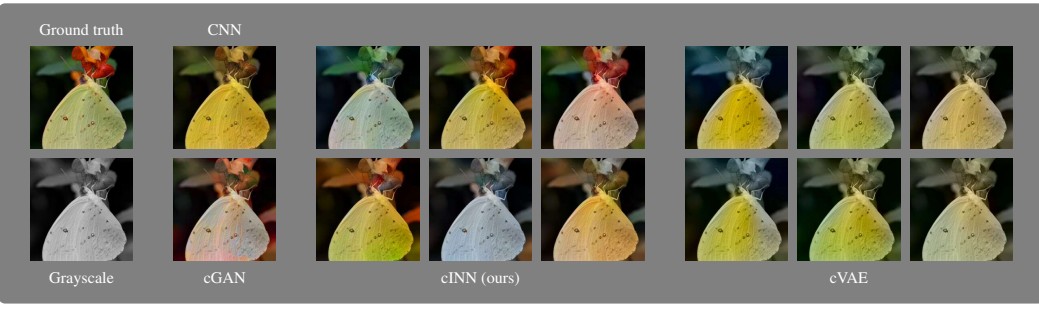

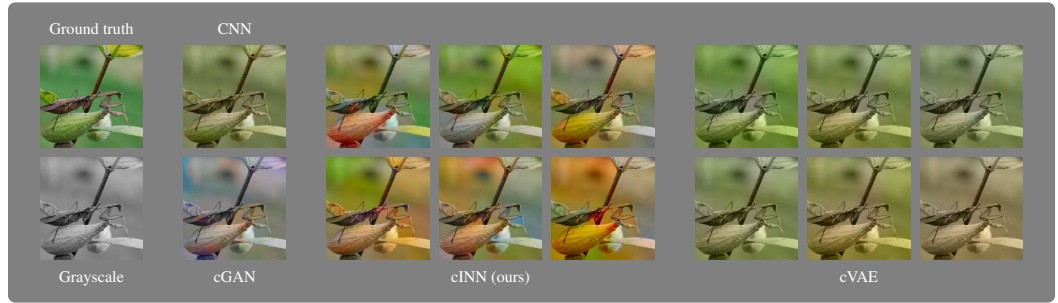

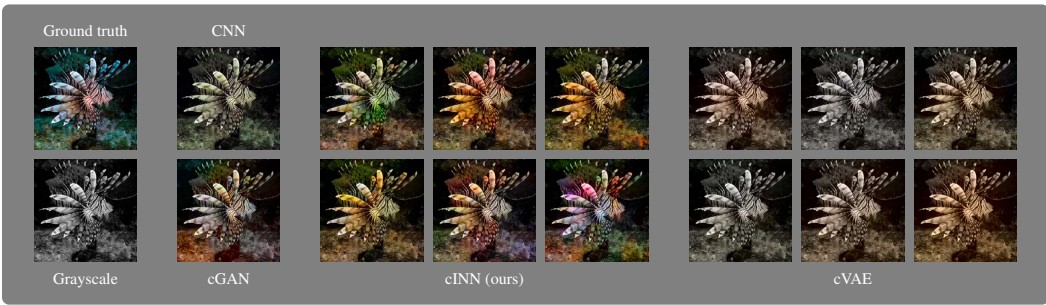

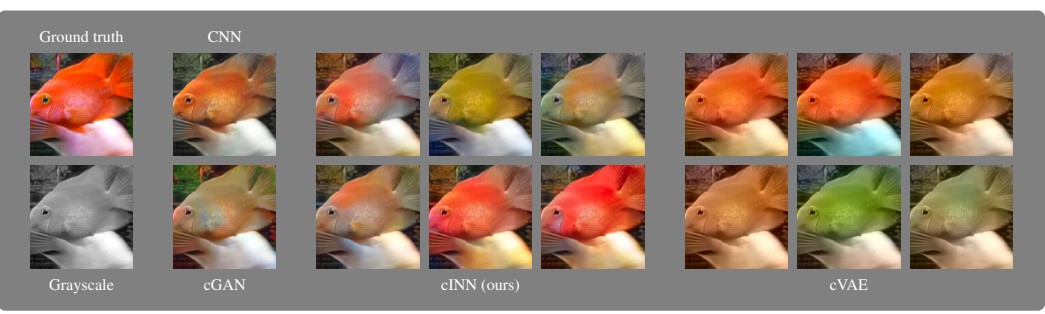

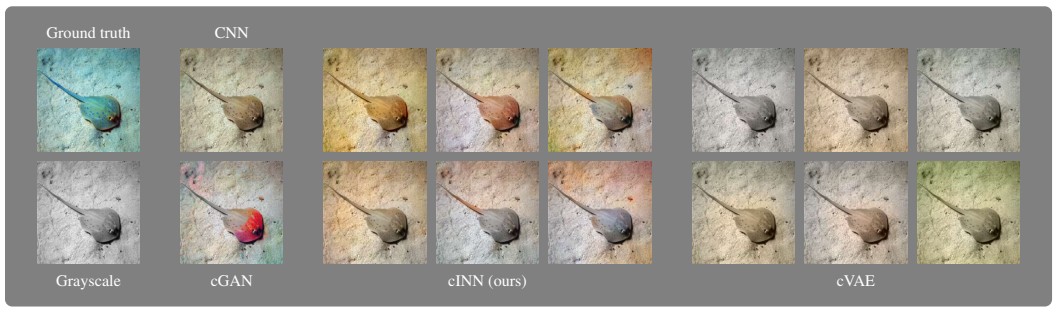

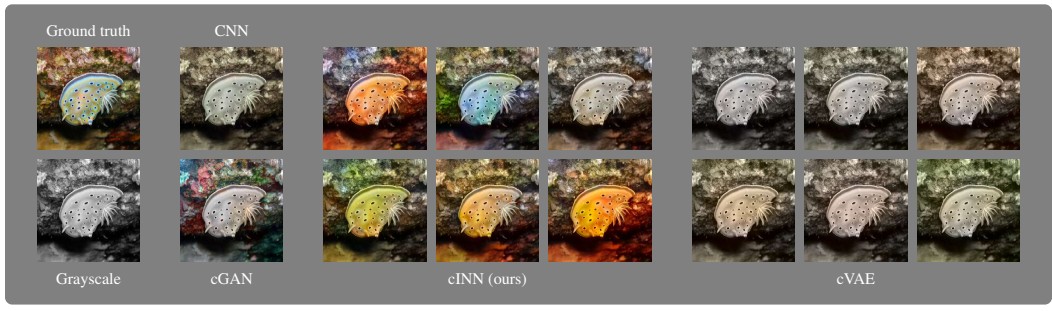

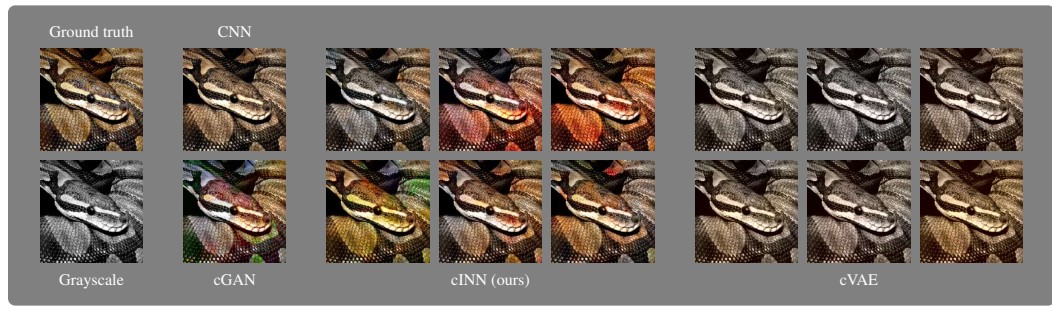

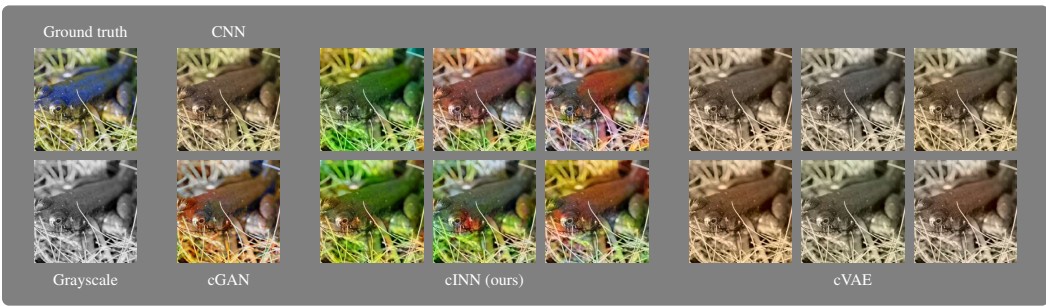

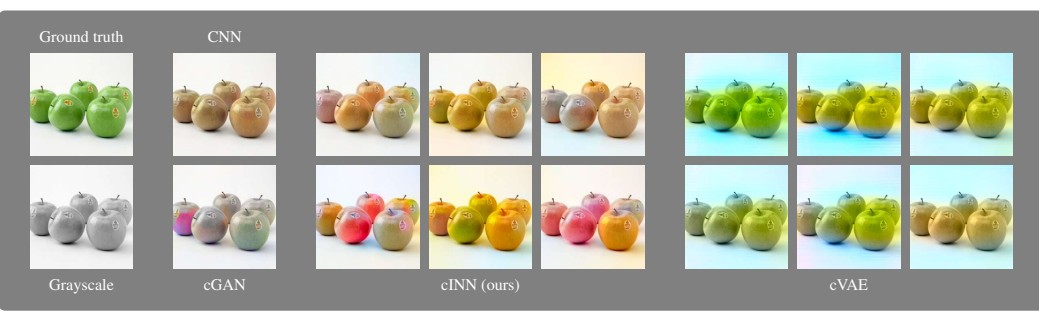

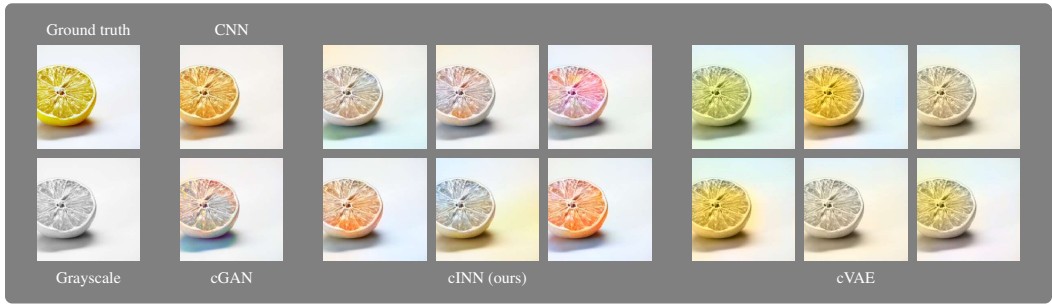

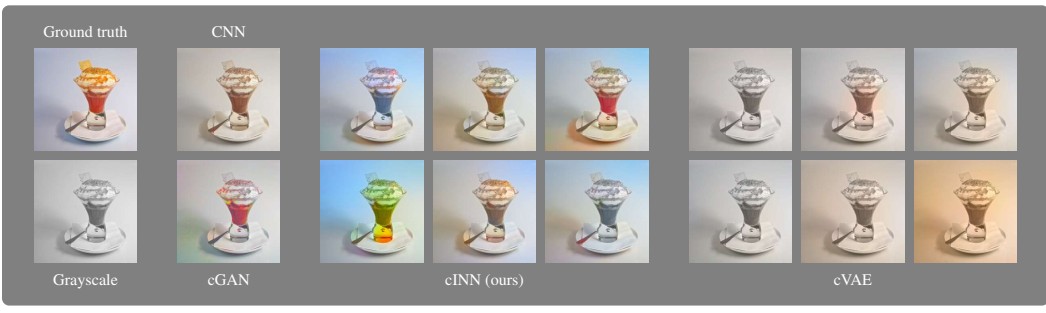

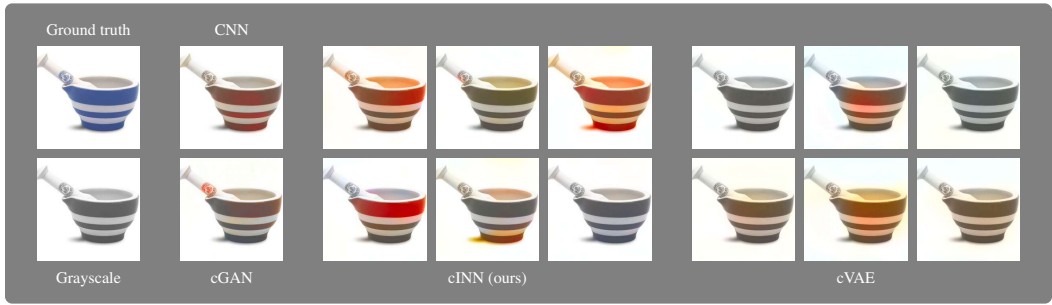

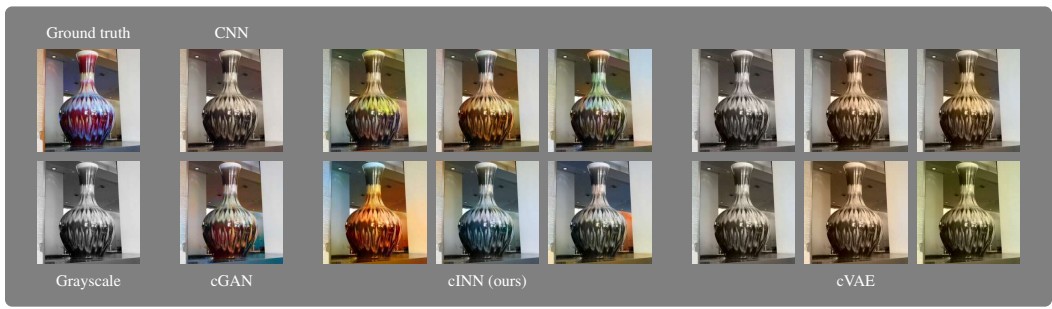

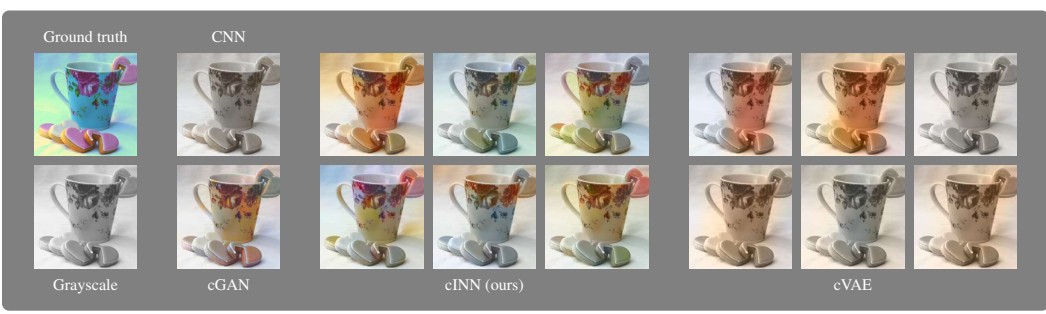

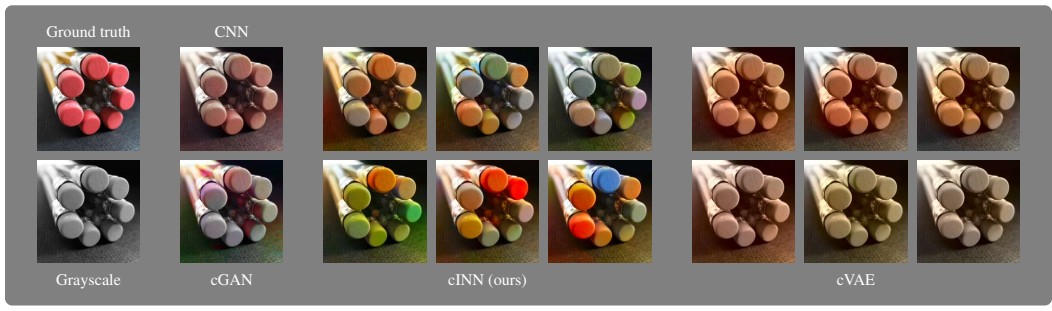

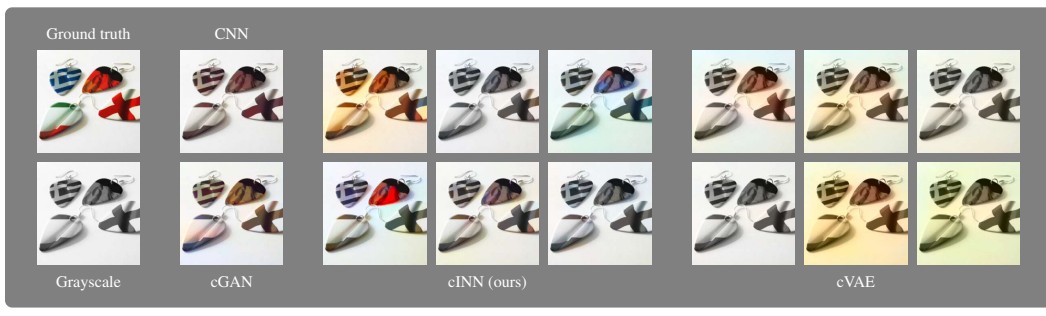

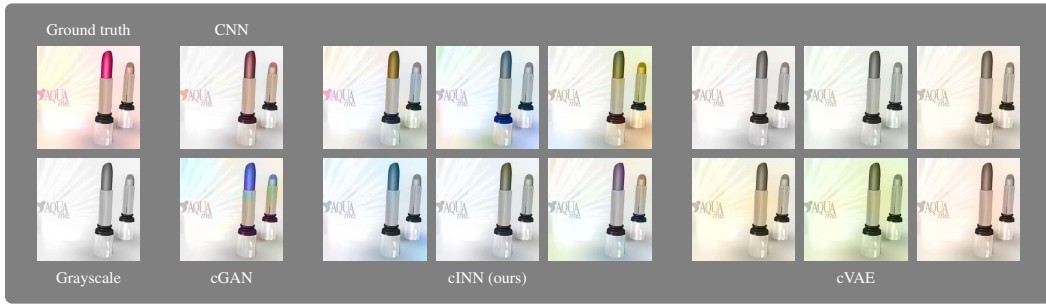

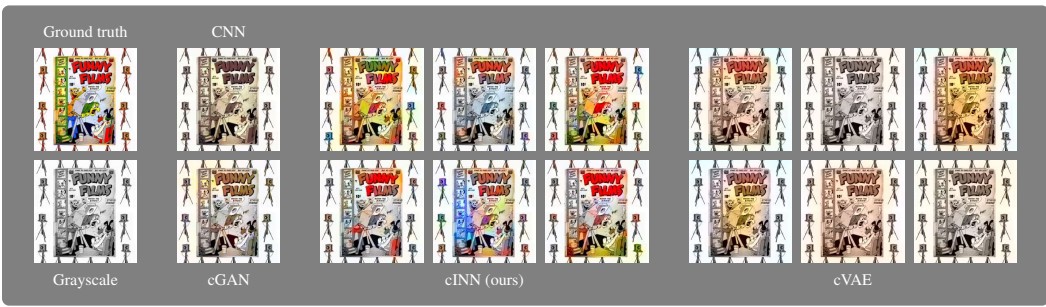

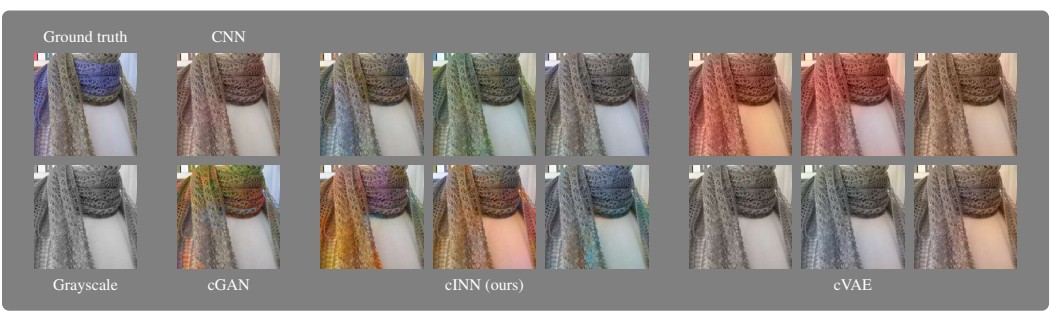

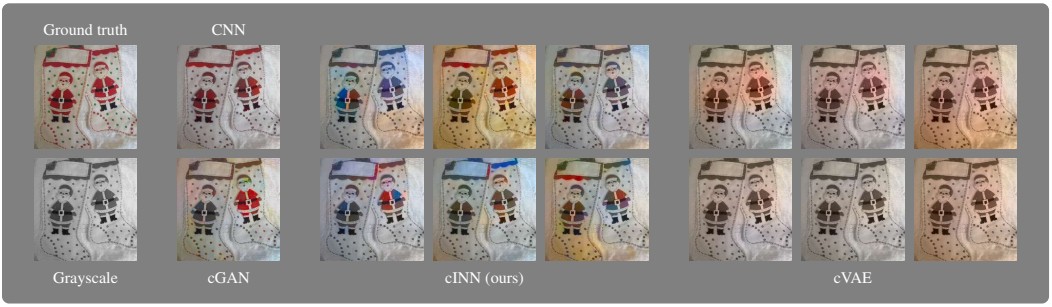

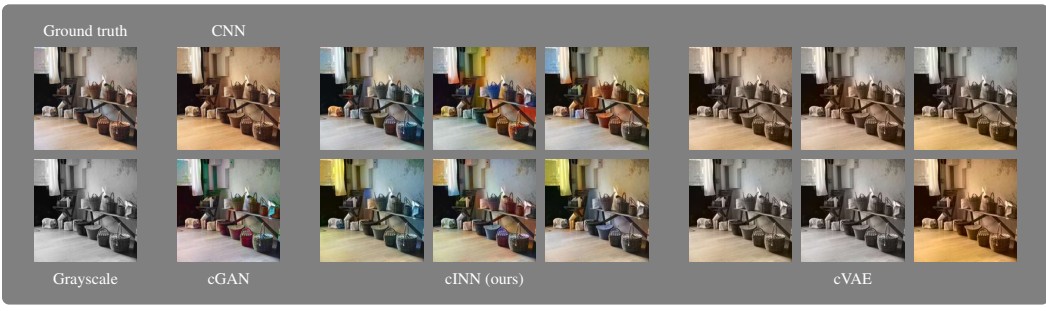

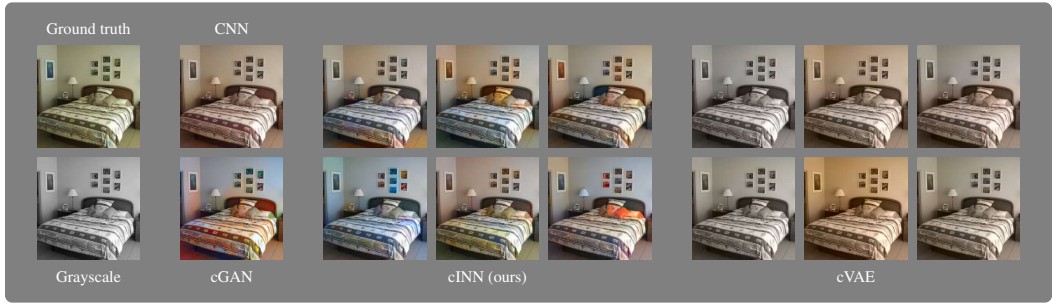

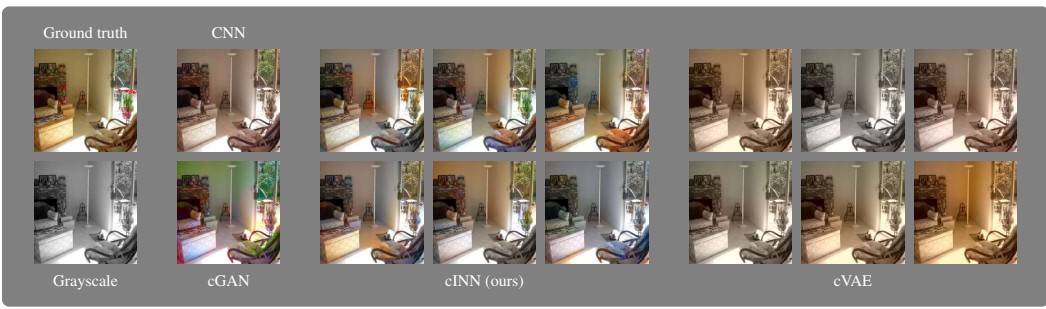

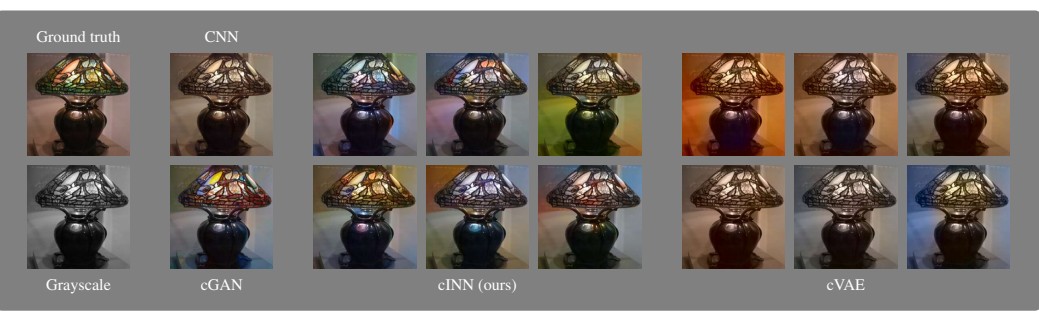

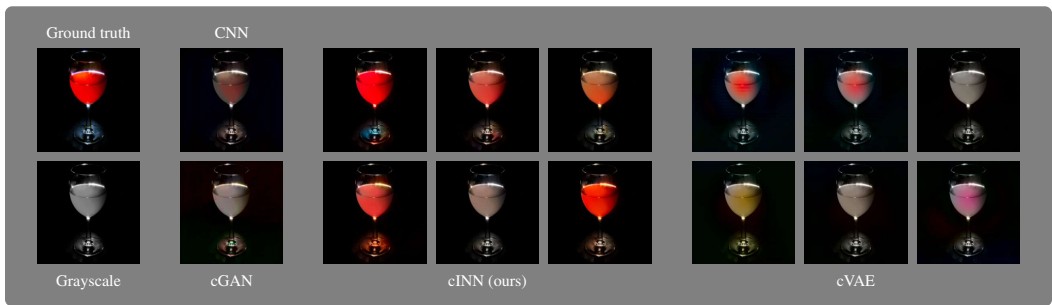

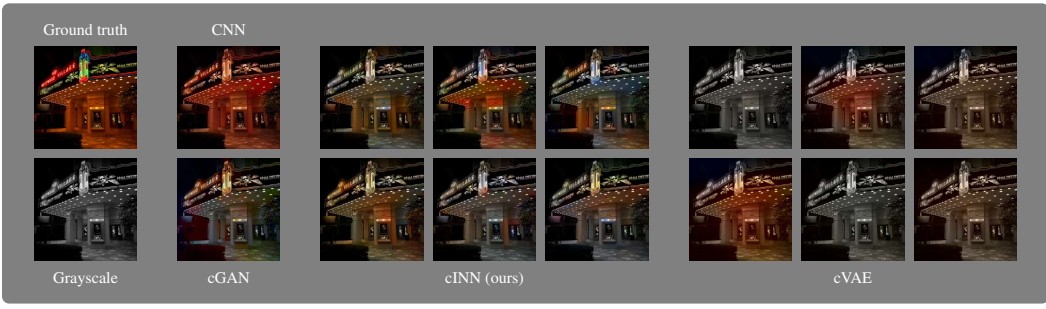

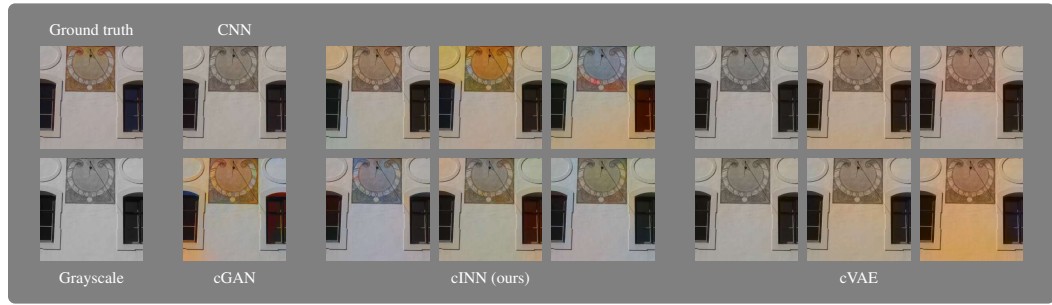

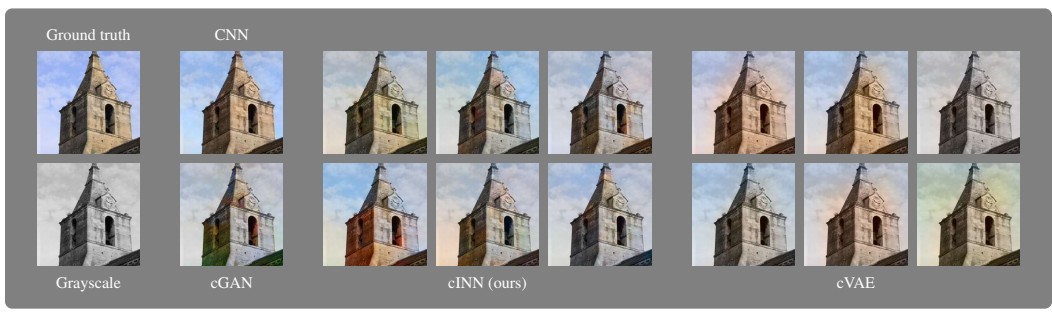

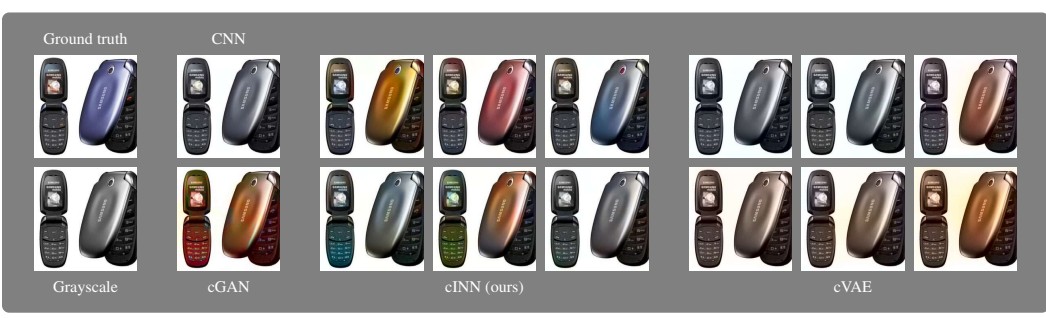

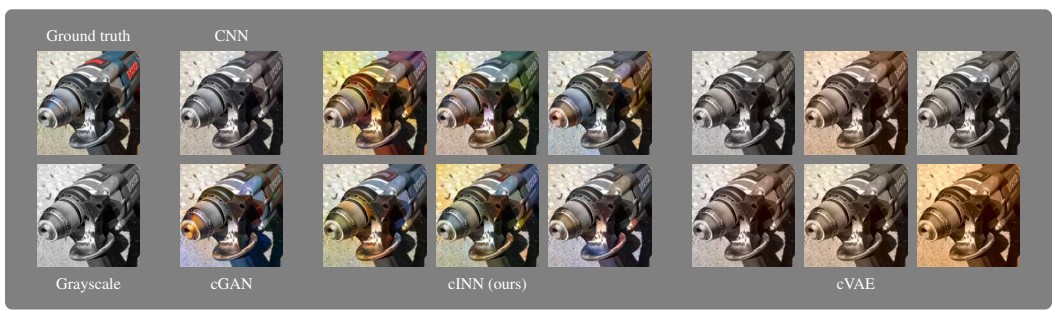

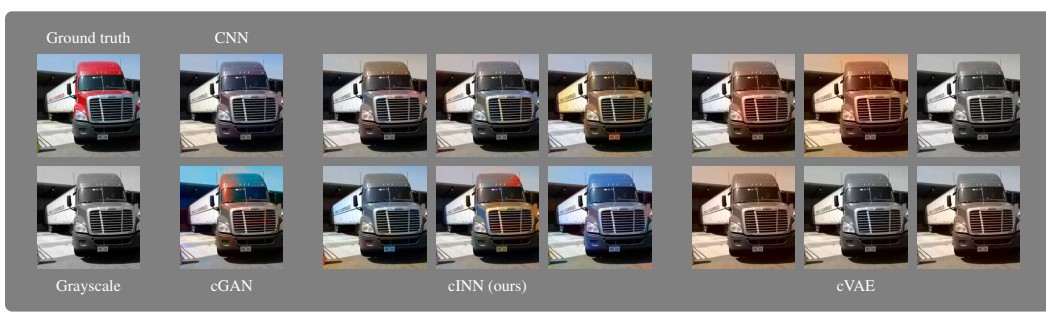

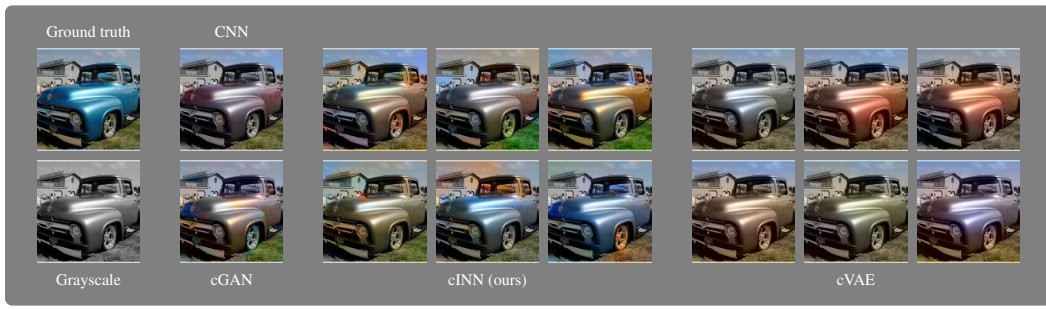

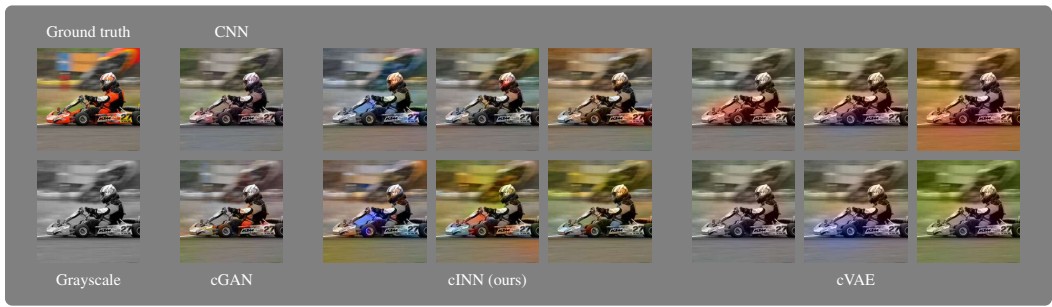

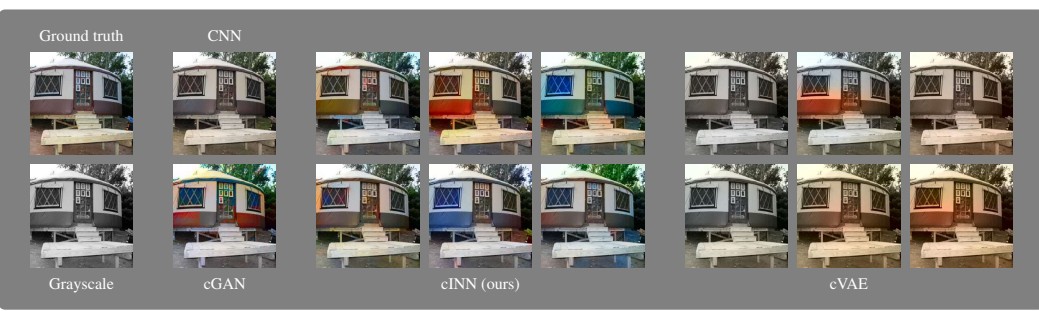

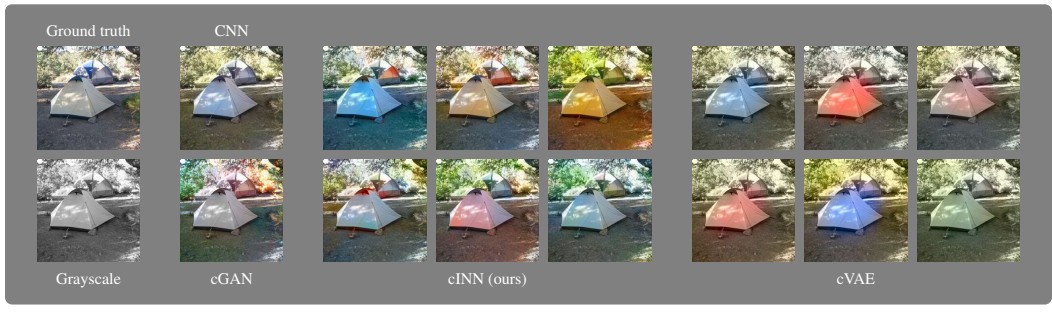

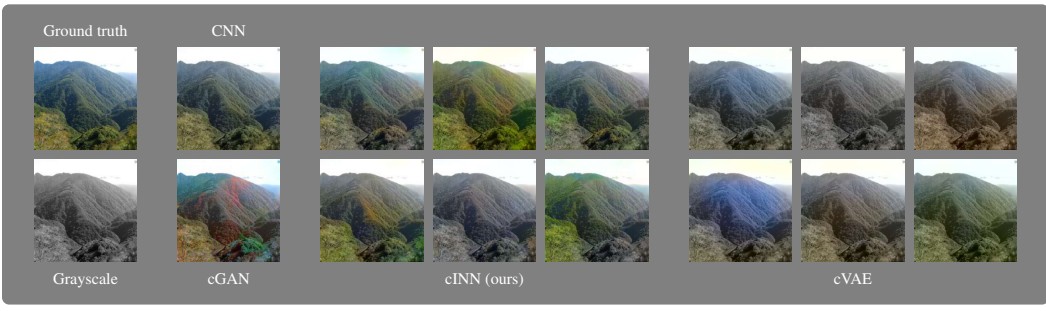

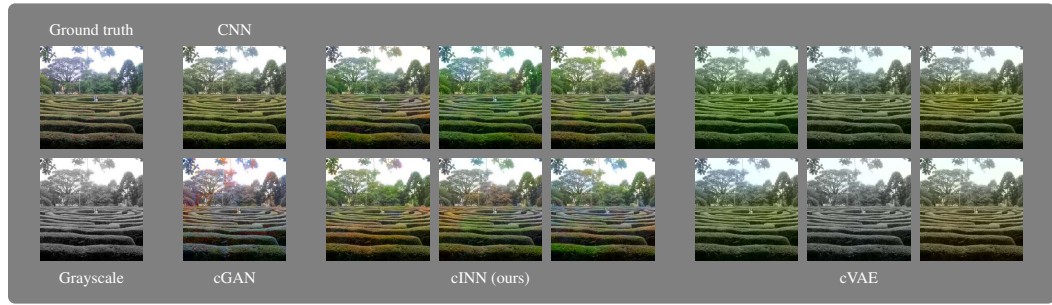

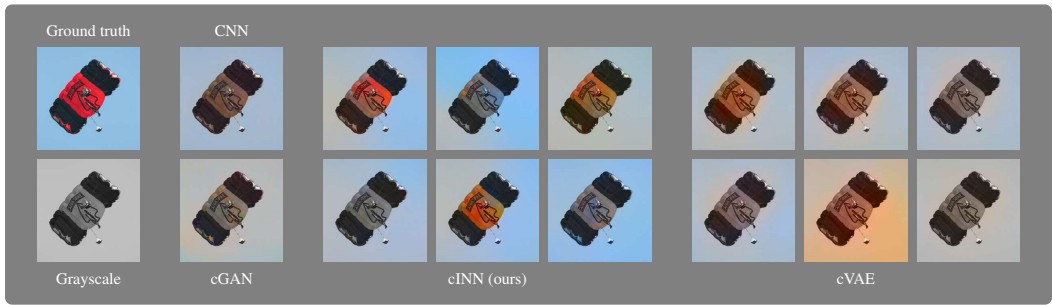

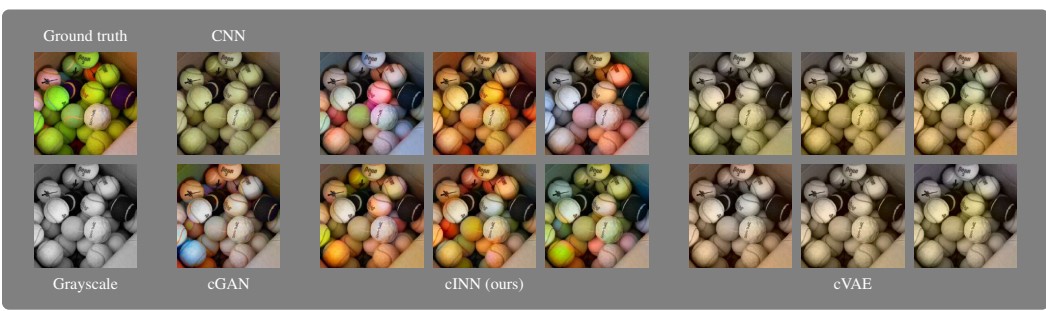

### B.3.2 HUMANS

We find that the cINN often has difficulties generating convincing skin colors, as shown below. Clothing is colored in diverse ways, but not always with the correct connectivity and consistency.

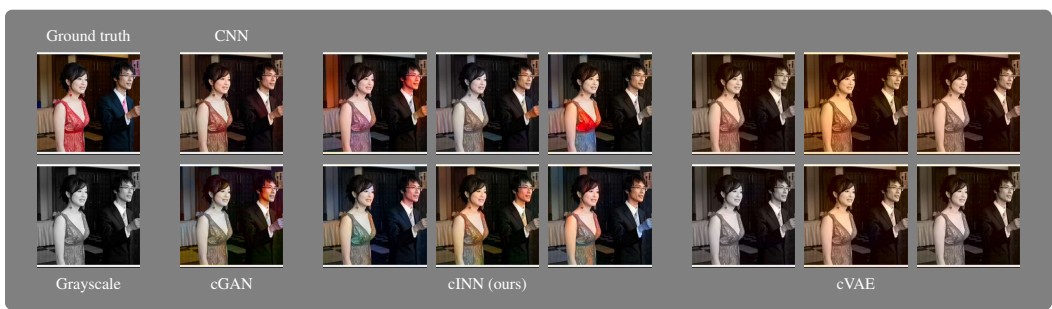

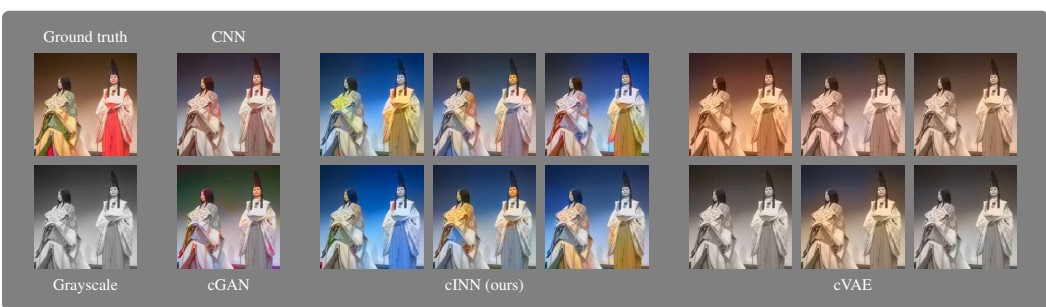

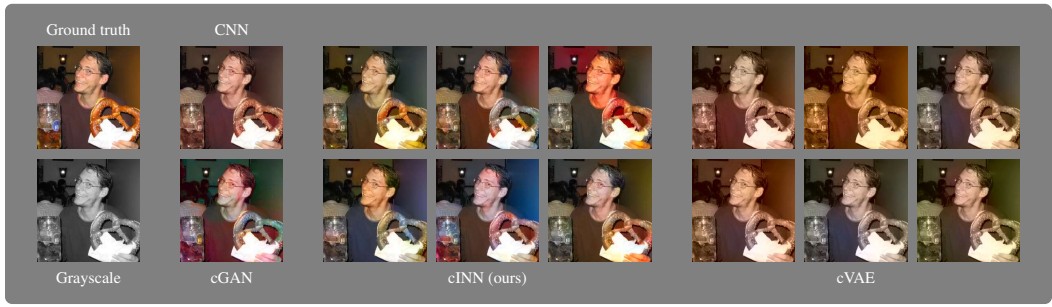

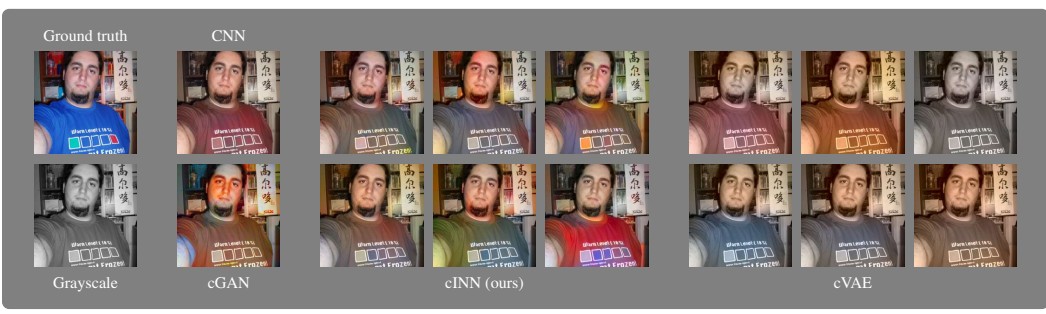

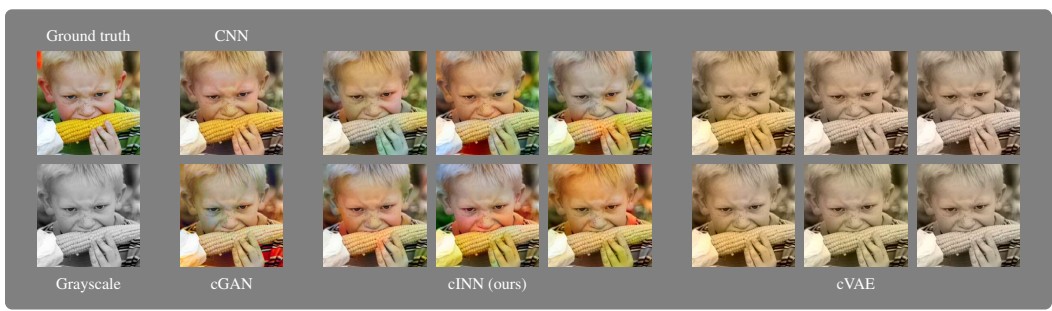

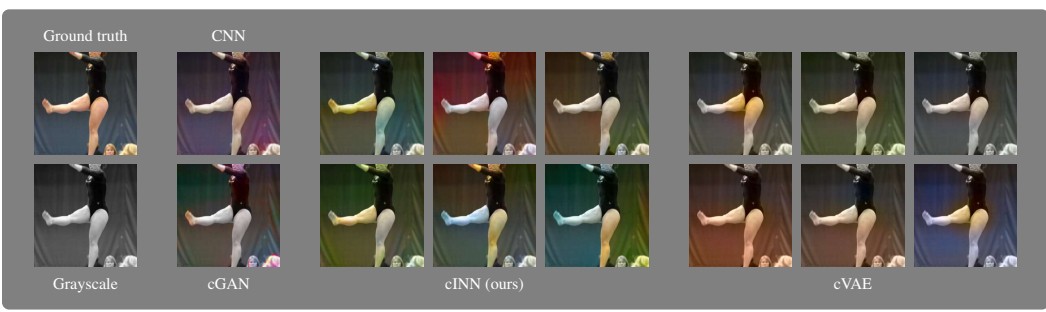

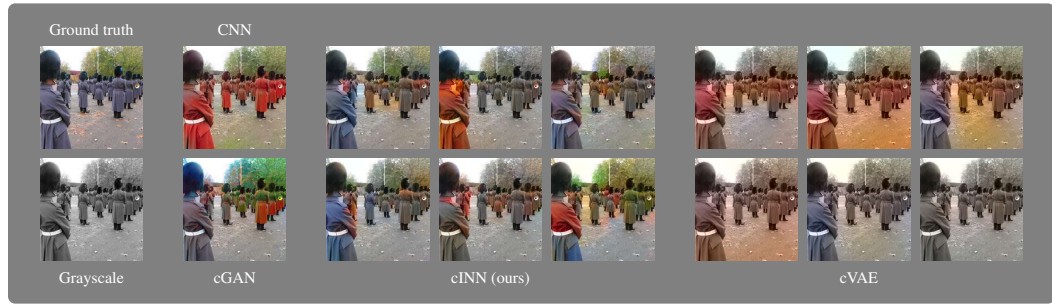

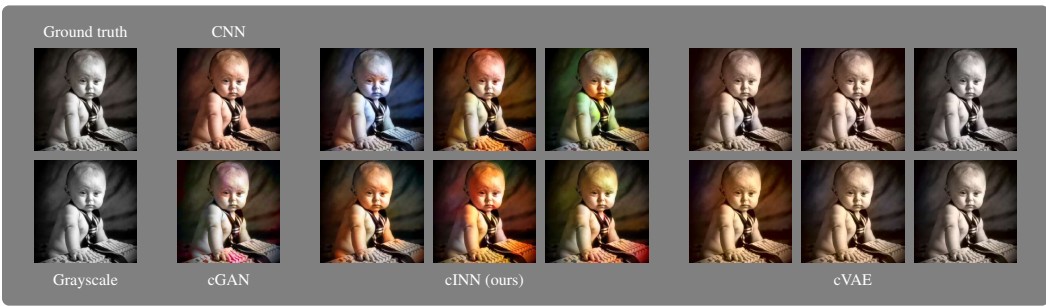

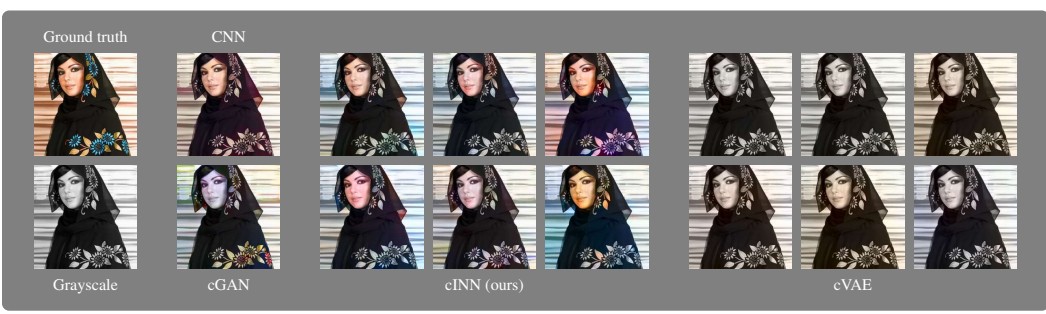

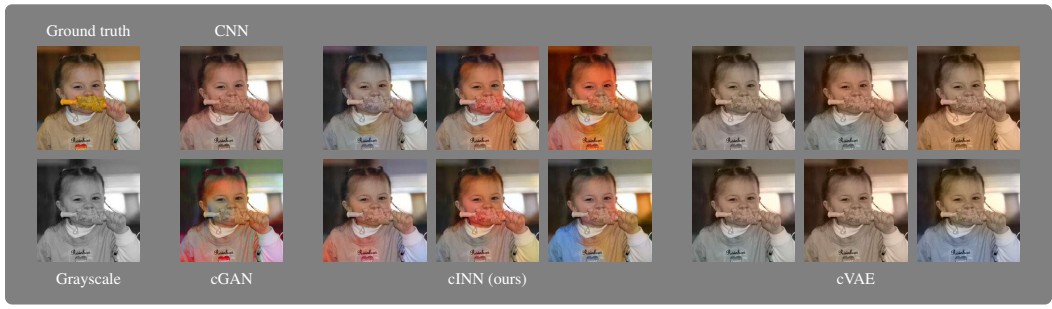

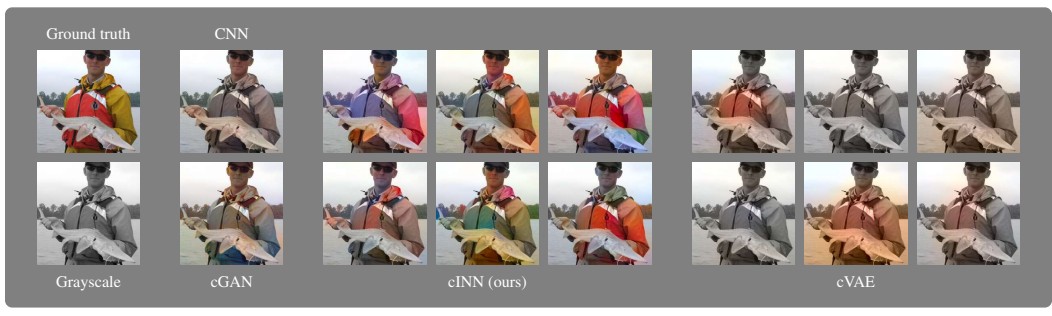

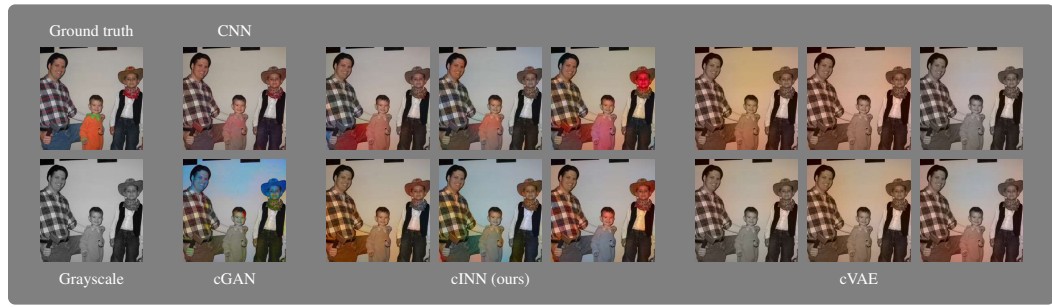

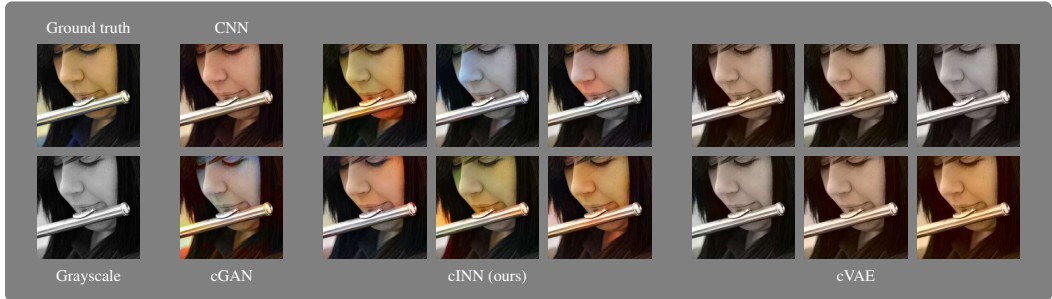

### B.3.3 LACKING CONSISTENCY

The following failure cases exhibit a lack in consistency, in occluded objects, multi-part objects, or reflections.

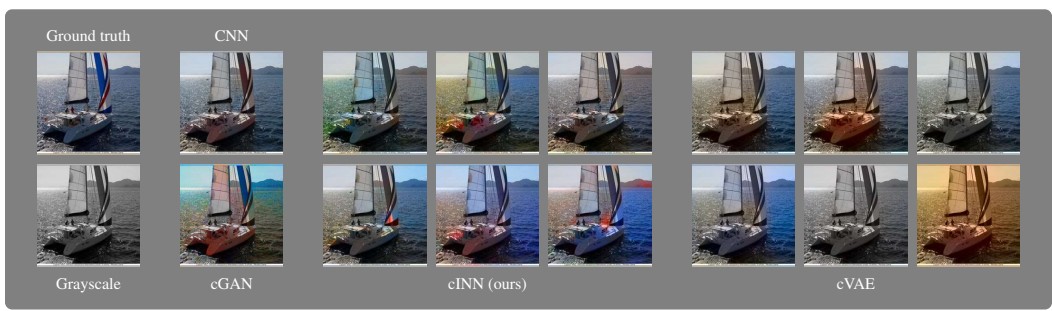

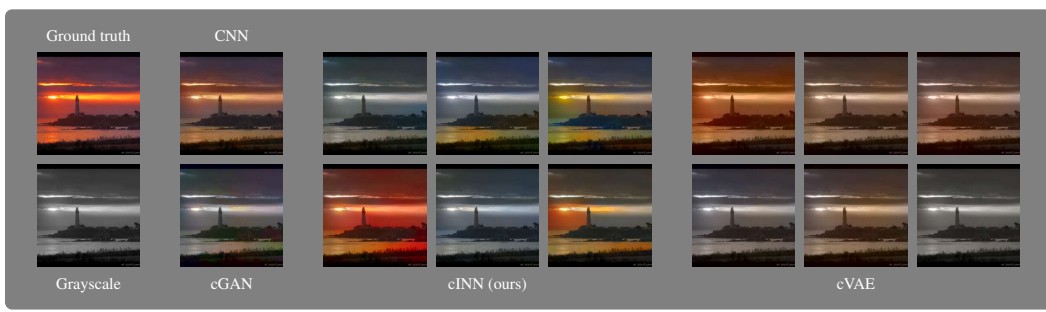

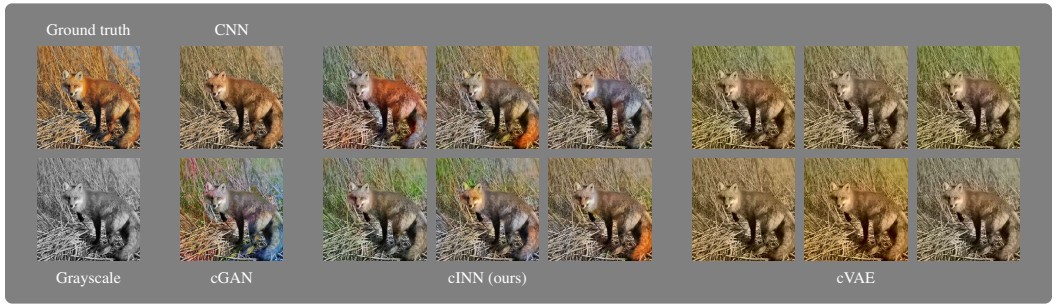

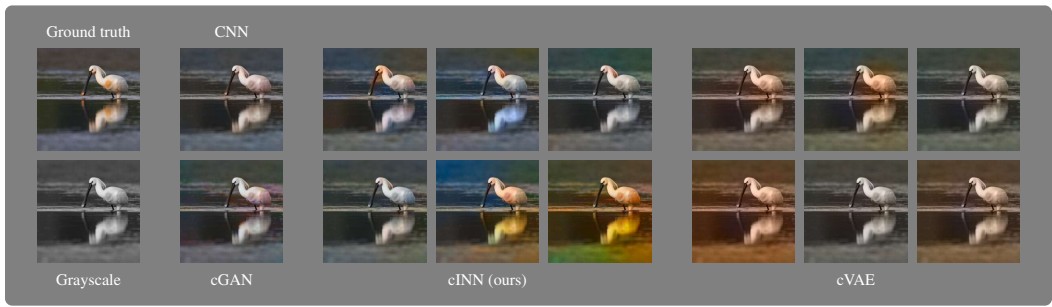

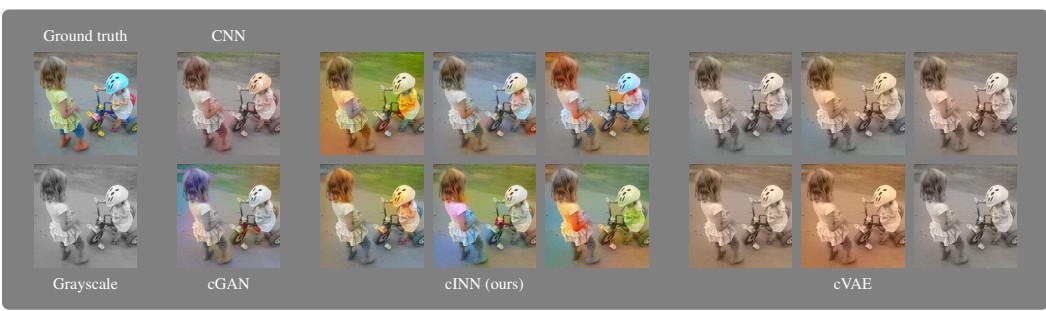

### B.3.4 Color ignores semantic content

In the following examples, the semantic content of the image was not recognized, and the generated colors are clearly incorrect.

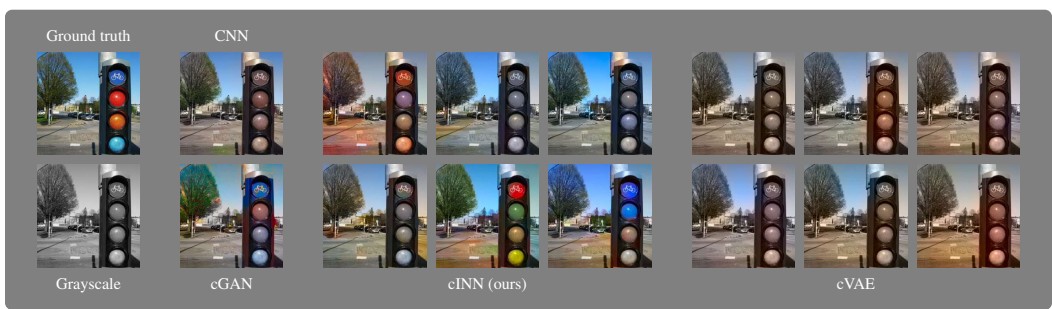

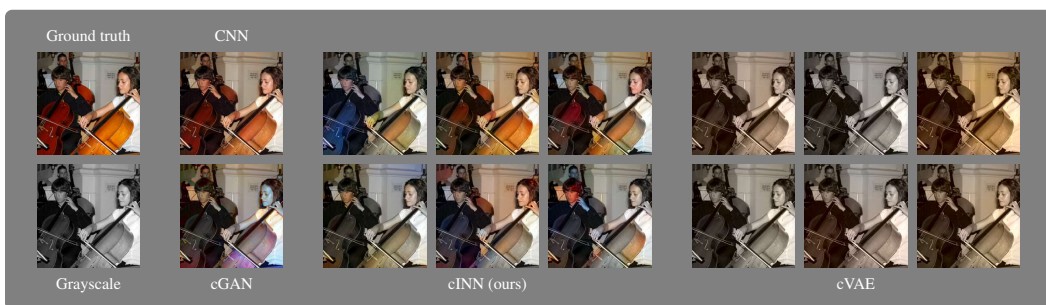

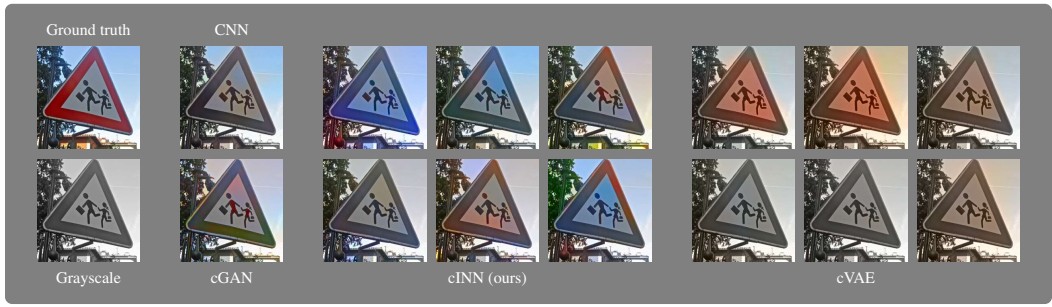

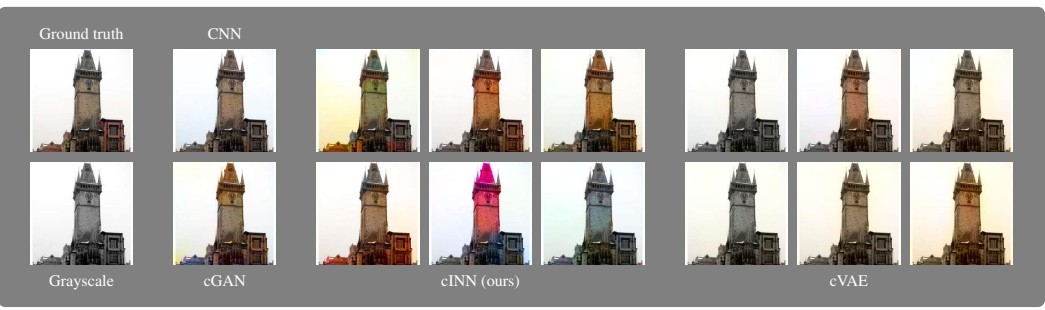

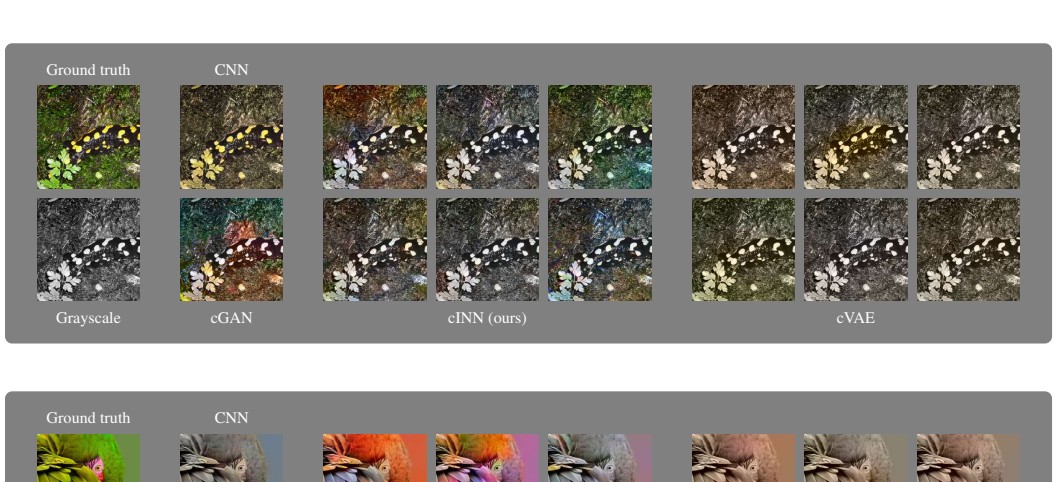

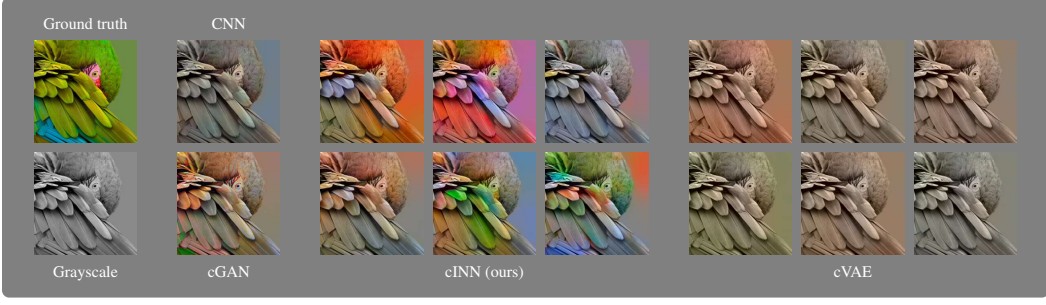

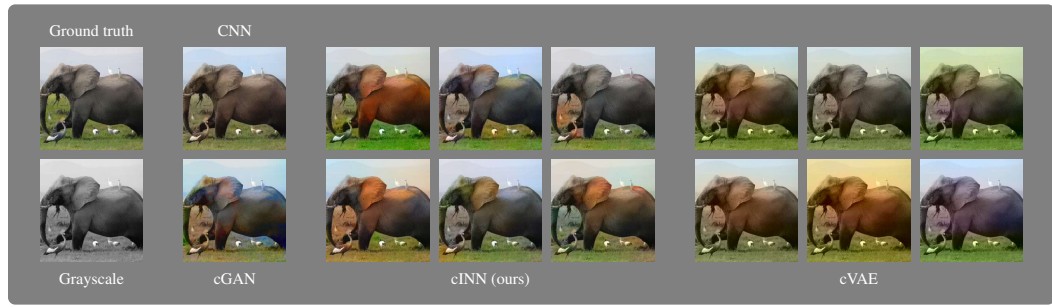

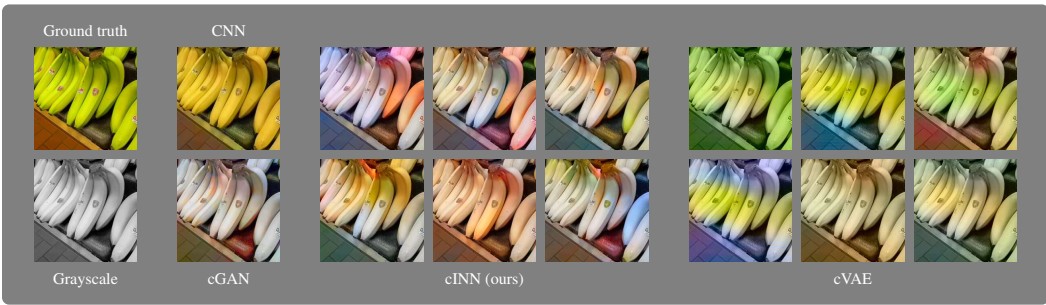

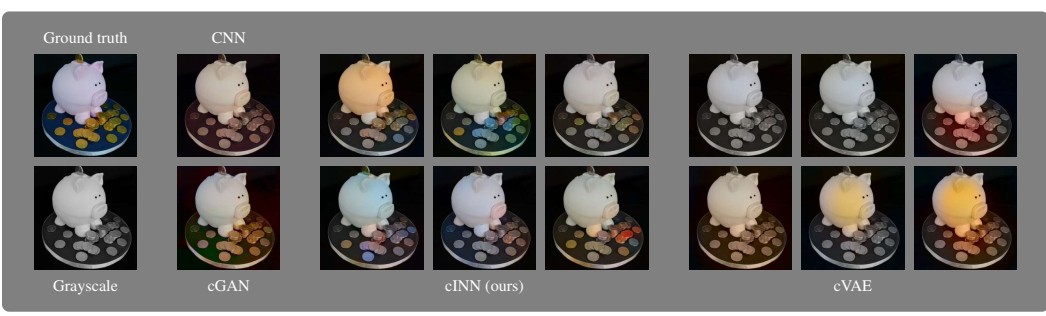

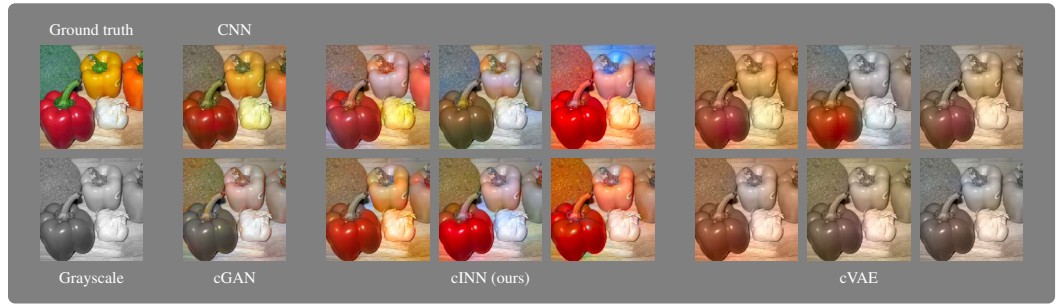

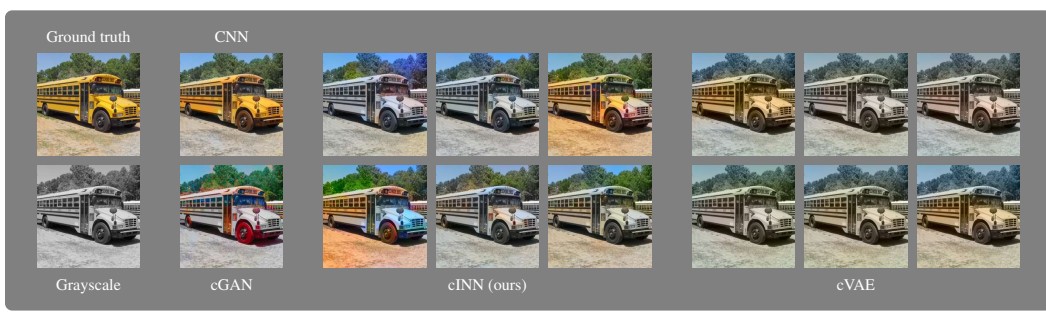

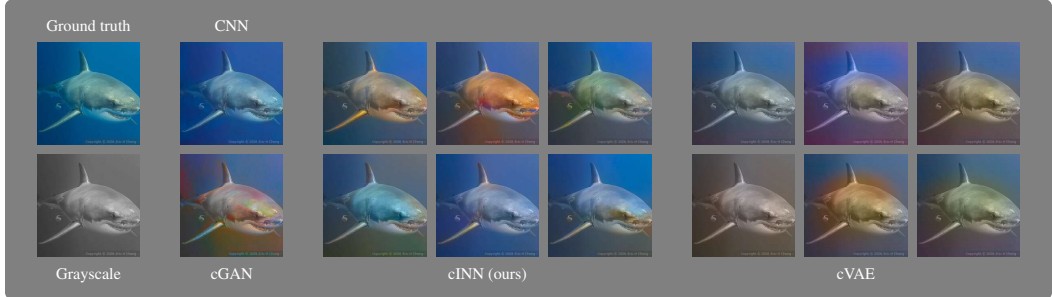

### B.3.5 Outright failures

For the following images, the cINN fails completely, and generates colors with seemingly little or no connection to the grayscale image.

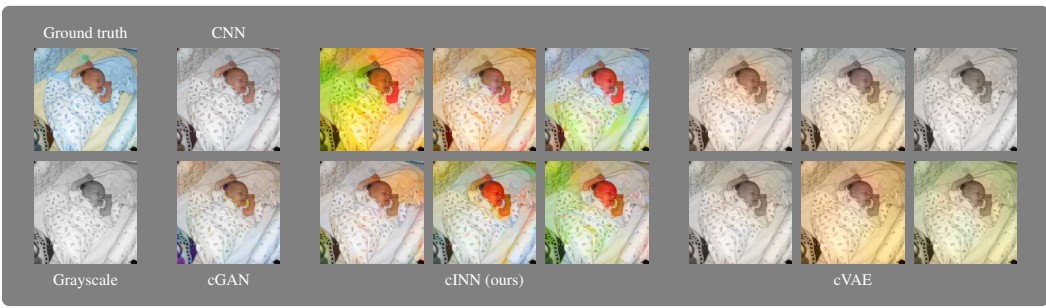

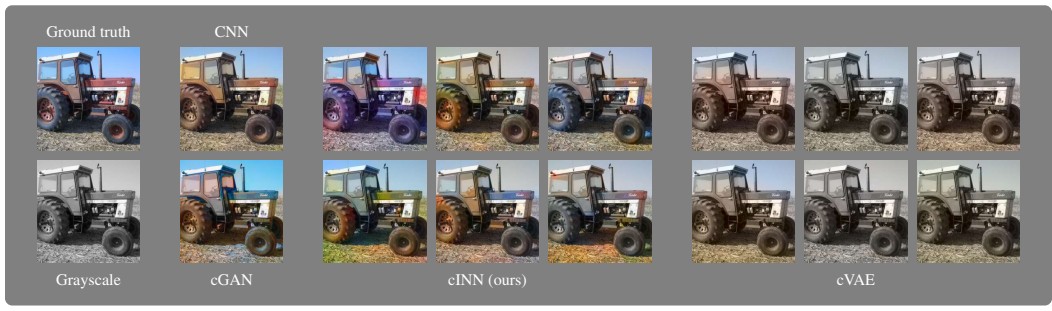

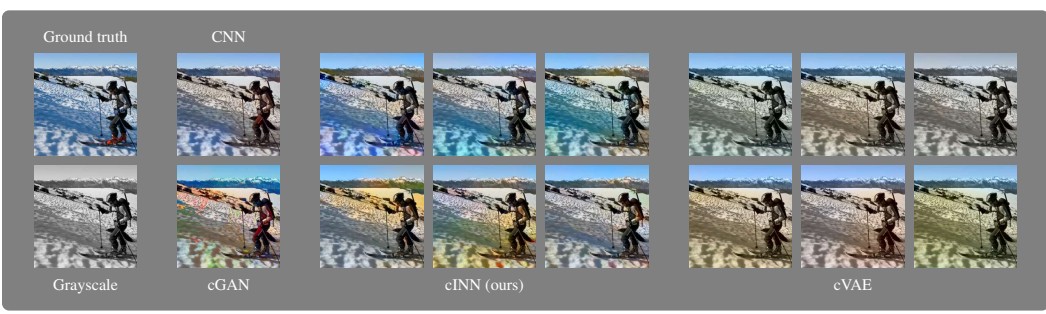

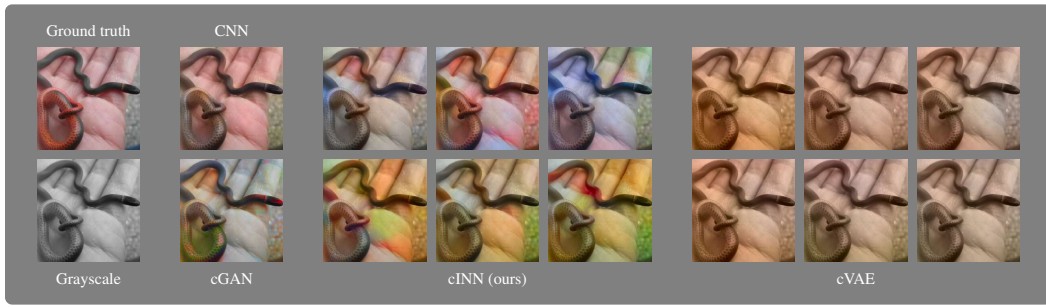

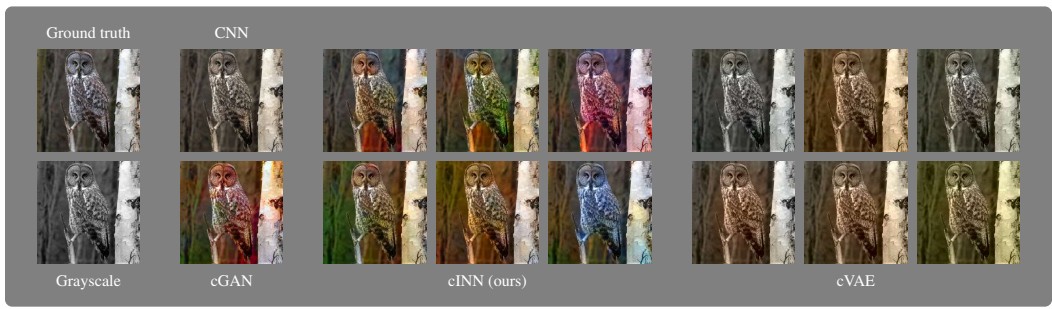

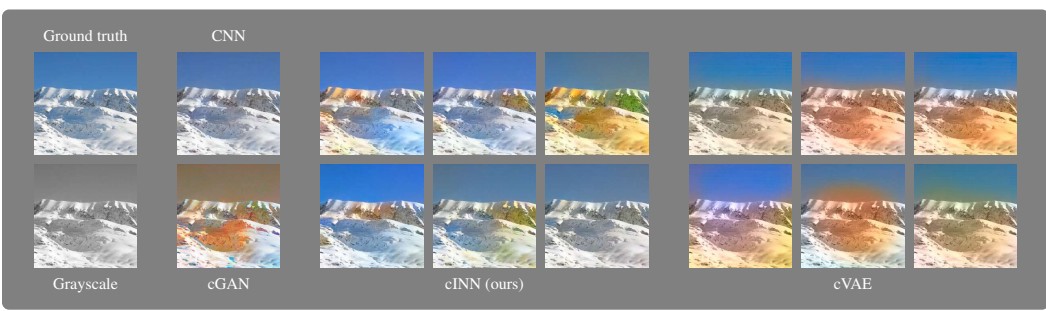

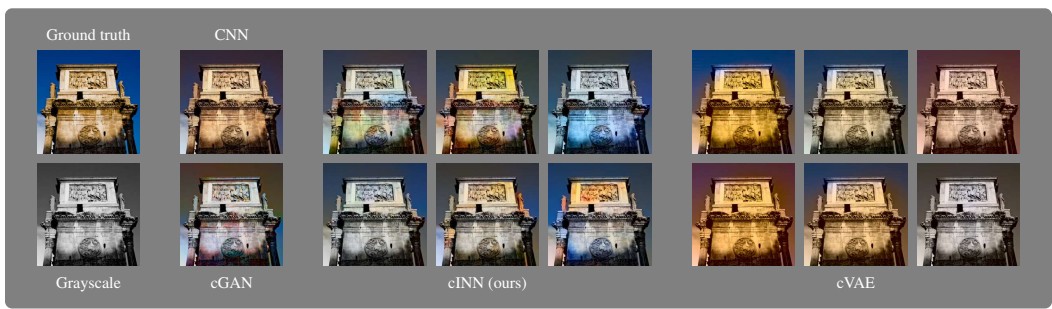

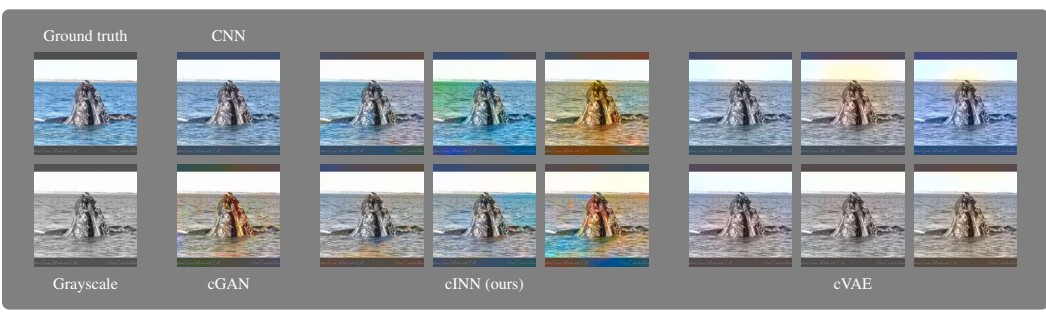

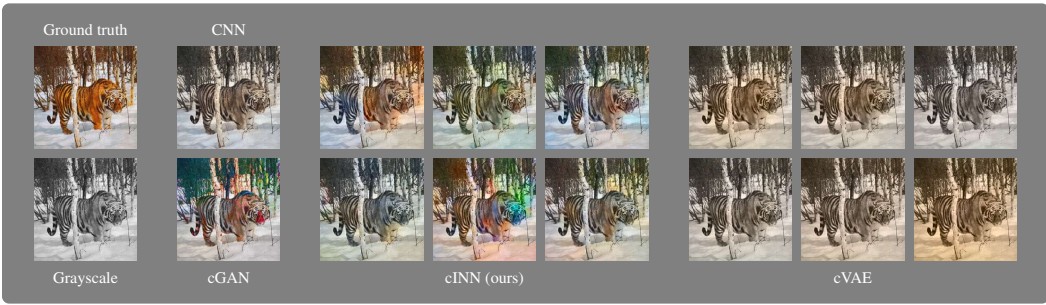

