# OpenReview forum: "Conditional Invertible Neural Networks for Guided Image Generation"
_ICLR.cc/2020/Conference — Reject_

### Official Review · AnonReviewer1 · 2019-10-20
**Official Blind Review #1**

**Rating:** 6

**Review:**

This paper proposes conditional Invertible Neural Networks (cINN), which introduces conditioning to conventional flow-based generative models. Conditioning is injected into the model via a conditional affine coupling block, which concatenates conditioning with the input to the scaling and shifting sub-networks in the coupling block. Other small modification are proposed to improve training stability at higher learning rates, including soft clamping of scaling coefficients, and Haar wavelet downsampling, which is proposed to replace the squeeze operation (pixel shuffle) that is often used in flow-based models. The invertibility of the cINN allows for style transfer by projecting images into the latent space, and then reconstructing the image with a different conditioning. The performance of the cINN is evaluated empirically on the task of colorization, where it is shown to outperform other techniques in terms of nearness to the true colours, as well as sample diversity.

Overall, I would tend to vote for accepting this paper. The base method for integrating conditioning into the flow is simple and intuitive, and additional modifications which allow for stable training at higher learning rates, such as soft clamping and Haar wavelet downsampling, appear to be very effective. Conditional models often lend themselves to a wide variety of useful applications, so I think this work could be of interest to many.


My primary concerns with this paper are related to the comparison of image colorization methods. Specifically:
1) I would like to see a comparison to Probabilistic Image Colorization (PIC) [1], which was mentioned in the related work section but not included in the comparison of colorization models. PIC has been shown to outperform VAE-MDN in terms of diversity, so it would be good to include it. Code is available online with pretrained ImageNet models (https://github.com/ameroyer/PIC), so it should not be difficult to add.

2) Pix2pix is known to have very bad sample diversity. A more useful comparison would be to evaluate one of the newer variants of Pix2pix that emphasizes sample diversity, such as BicycleGAN [2] or MSGAN [3]. Code is also available for each of these models (https://github.com/junyanz/BicycleGAN, https://github.com/HelenMao/MSGAN), although you would need to train models from scratch.

3) Pixel-wise metrics such as MSE are bad at measuring perceptual similarity. While these pixel-wise metrics are still useful for comparison to prior work, better metrics are available for evaluating image colorization. I would recommend the use of Learned Perceptual Image Patch Similarity (LPIPS) [4] in place of pixel-wise distance measures for evaluating image similarity and diversity.


Things to improve the paper that did not impact the score:
5) I was somewhat disappointed by how little attention was spent on the Haar wavelet downsampling method. It seems like a very neat idea, but it is only briefly explored in the ablation study. It would be nice to include a more in-depth study of how it compares to the conventional pixel shuffle downsampling, perhaps in terms of stability with learning rates, and final model performance.

6) There is some potentially related work on conditional adversarial generative flows [5] that could be added to the literature review if deemed relevant enough.


References:
[1] Amelie Royer, Alexander Kolesnikov, and Christoph H. Lampert. Probabilistic image colorization. In British Machine Vision Conference (BMVC), 2017.

[2] Zhu, Jun-Yan, et al. "Unpaired image-to-image translation using cycle-consistent adversarial networks." Proceedings of the IEEE international conference on computer vision. 2017.

[3] Mao, Qi, et al. "Mode seeking generative adversarial networks for diverse image synthesis." Proceedings of the IEEE Conference on Computer Vision and Pattern Recognition. 2019.

[4] Zhang, Richard, et al. "The unreasonable effectiveness of deep features as a perceptual metric." Proceedings of the IEEE Conference on Computer Vision and Pattern Recognition. 2018.

[5] Liu, Rui, et al. "Conditional Adversarial Generative Flow for Controllable Image Synthesis." Proceedings of the IEEE Conference on Computer Vision and Pattern Recognition. 2019.

### Post-Rebuttal Comments ###

Thank you for including the PIC results. I did not realize that they would take so long to run, but I think it important to include comparison with the current SOTA methods so that we have some reference. It is also nice to see the addition of the LPIPS metric. My overall opinion of the paper is not much changed from before, so I will retain the same score.


**Experience Assessment:**

I have published one or two papers in this area.

**Review Assessment: Checking Correctness Of Derivations And Theory:**

N/A

**Review Assessment: Checking Correctness Of Experiments:**

I carefully checked the experiments.

**Review Assessment: Thoroughness In Paper Reading:**

I read the paper at least twice and used my best judgement in assessing the paper.

---

> ### Author Response · Authors · 2019-11-14
> **Review Response**
>
> Thank you for the valuable comments, which helped us improving our paper a lot.
> We address each point below. We also updated the PDF, highlighting all changes in red.
>
> 1.) We agree that the comparison to PIC is a useful addition. Due to slow inference, PIC was not yet able to finish processing the entire test set. Partial results (included in our revised paper) indicate that PIC slightly outperforms the cINN in some metrics, with two major differences:
> a) The autoregressive approach employed by PIC is extremely time consuming. Using the original authors’ code, colorizing a batch of 16 images takes 40 minutes on an RTX 2080Ti GPU. Our cINN finishes in the order of milliseconds for the same images.
> b) Latent space manipulations and style-transfer were not demonstrated in the PIC paper and may not be straight-forward in an autoregressive model.
>
> 2.) We agree that comparisons to more recent GANs would be very interesting.
> However, Bicycle GANs and MSGANs have not yet been applied to colorization, and adapting their architectures, training, and hyperparameters to this task is beyond the scope of our paper. Instead, we compare to the COLORGAN, which also claims to achieve diverse colorization. The comparison to COLORGAN is briefly discussed in the text, and more detailed results can be found in the appendix.
>
> 3.) This is a useful suggestion, we have included the LPIPS metric in the quantitative results. The measurements seem to reaffirm the results so far, with the exception that the VAE-MDN performs much worse in LPIPS-diversity than RGB-diversity.
>
> 5.) We have added more motivation for and explanations of Haar wavelet downsampling. Deeper investigations and comparisons with alternative spatial transforms are indeed very interesting, but will be left for future work.
>
> 6.) Thank you for bringing this reference to our attention. We have added it to the related work section.

---

### Official Review · AnonReviewer3 · 2019-10-23
**Official Blind Review #3**

**Rating:** 6

**Review:**

The paper presents an invertible generative network, for conditional image generation.  The model is an extension of Real NVP with a conditioning component. Experiments are performed for image generation on two tasks: class conditional generation on MNIST and image colorization conditioned on a grey scale image (luminance). Comparisons are performed with a conditional VAE and a conditional GAN (Pix2Pix). An ablation study motivates the importance and role of the different components.
The model itself is a relatively simple extension of Real NVP, where a condition vector is added to the initial model as an additional input to the NN components of the invertible blocks. In the experiments conditioning may be a simple class indicator (MNIST) or a more complex component corresponding to a NN mapping of an initial conditioning image (colorization). The experiments show that this model is able to generate good quality images, with an important diversity, showing that the conditioning mechanism works well. The quantitative comparison also shows that the proposed model is competitive with two baselines taken in the VAE and GAN families. The model works well for the non-trivial task of colorization.
The authors claim is that they are the first to propose conditional invertible networks. The main contribution is probably the implementation of the model itself. They make use of several “tricks” that improve a lot on the performance as demonstrated by the ablation study.  As such more details and motivations for these different ideas that improve the performance and stability of the model would be greatly helpful.  It looks like these are not details, but requirements to make the whole thing work. The Haar component for example should be better motivated. There is no comparison in the ablation study with an alternative, simpler decomposition.
The baselines are probably not the strongest models to date, and better results could certainly be obtained with other VAE or GAN variants. For example, there have been several works trying to introduce diversity for GANs. This is not redhibitory, but this should be mentioned. Besides a short description of the two baselines, would make the paper more self-contained.
The quantitative comparison with the VAE baseline, shows that the two models are quite similar w.r.t. different measures. This could be also commented.
The notations for the Jacobian do not integrate the conditioning, this could be corrected.
Concerning the interpretation of the axis for the MNIST experiment, it is not clear if they are axis in the original space or PCA axis. If this is the first option, more details are needed in order to understand how they were selected.


------post rebuttal -----

The authors clarified several of the raised points. I keep my score.



**Experience Assessment:**

I have read many papers in this area.

**Review Assessment: Checking Correctness Of Derivations And Theory:**

I assessed the sensibility of the derivations and theory.

**Review Assessment: Checking Correctness Of Experiments:**

I assessed the sensibility of the experiments.

**Review Assessment: Thoroughness In Paper Reading:**

I read the paper thoroughly.

---

> ### Author Response · Authors · 2019-11-14
> **Review Response**
>
> Thank you for your review, and the valuable comments.
> We address your main points below. We also updated the PDF with the suggestions, highlighting changes in red.
>
> Points concerning clarity/completeness of the text:
> We have revised sections that may have been unclear or too brief:
> - Clearer motivation for Haar wavelet downsampling
> - Better notation of the Jacobian
> - Clarification how the disentangled style axes were found with PCA
>
> Concerning the baselines:
> Colorization is not a standard benchmark for generative models, and the most recent architectures have not been applied to this task so far. In fact, the state-of-the-art is still defined by regression-based models. We have compared our method with all existing models (both regression and generative) that were specifically proposed for colorization. In particular, we added Probabilistic Image Colorization (PIC) to the revised version of our paper. More recent generative architectures certainly have the potential to achieve better diversity than Isola et al. (2017), but adapting, tuning, and training these models for colorization is beyond the scope of our paper.
>
> A baseline alternative for Haar wavelet downsampling is the established pixel rearrangement operation found in iRevNet, Real-NVP and Glow, which we did in fact compare to in our ablation study (Fig. 15, right) under the heading “No Haar”. As can be seen there, our model did not converge well under this downsampling strategy.

---

### Official Review · AnonReviewer2 · 2019-11-04
**Official Blind Review #2**

**Rating:** 3

**Review:**

Authors provide an extension to the invertible generative models, by fashioning them into conditional invertible generative networks. The networks are conditioned at the input with a feed-forward network initialized with a VGG based classification network. Conditioning is implemented within the coupling layers (Dinh et. al. 2016) of the invertible model by simply concatenating the hidden layer output of the VGG encoder. The model is learned using an MAP objective along with some modifications to the original training procedure (described in sec 3.4). The model is evaluated qualitatively on "style transfer" on MNIST digits and image colorization. The technical contribution of this paper is the somewhat straight-forward extension of the cINNs to conditional generative networks. The actual implementation of conditioning seems quite trivial (sec 3.1). Although the results on colorization are claimed to be good, the baselines they compared to are not very recent (e.g. cGANs). Overall, I believe there is very less novelty, technical sophistication and performance improvements in this paper.

**Experience Assessment:**

I have read many papers in this area.

**Review Assessment: Checking Correctness Of Derivations And Theory:**

I assessed the sensibility of the derivations and theory.

**Review Assessment: Checking Correctness Of Experiments:**

I assessed the sensibility of the experiments.

**Review Assessment: Thoroughness In Paper Reading:**

I read the paper thoroughly.

---

> ### Author Response · Authors · 2019-11-14
> **Review Response**
>
> Thank you for taking the time to review our paper and for your feedback.
> We address your main points below. We also updated the PDF with suggestions from all reviewers, highlighting changes in red.
>
> Concerning the straight-forward nature of the architecture extension:
> As stated in our claims, our improvements to stabilize maximum likelihood training are crucial for making cINNs work in a challenging real-world computer vision task.
> To the best of our knowledge, this has not previously been demonstrated with conditional invertible architectures. While our contributions are somewhat technical, they have the potential to enable major new applications and interesting follow-up work.
>
> Concerning the baselines:
> Colorization is not yet among the standard benchmarks for generative models. In fact, the state-of-the-art in this area is defined by regression-based models. We have compared our method with all existing models (both regression and generative) that were specifically proposed for colorization. Adapting, tuning, and training more recent generative architectures for this task, although certainly promising, is beyond the scope of our paper.

---

### Official Review · AnonReviewer4 · 2019-11-04
**Official Blind Review #4**

**Rating:** 8

**Review:**

The authors propose to use a normalizing flow architecture to tackle the structured output problem of generalization.
They propose:
- a conditioning architecture: they use a convolutional feature extractor (similar to a U-Net architecture), and (on top of the common architectural details of models like Glow - Kingma and Dhariwal, 2018) uses Haar wavelets for downsampling;
- they train their architecture stably using the maximum likelihood principle;
- they demonstrate interesting properties of their model coming from the bijectivity.

This is an interesting application of the architecture to colorization. The diverse and consistent colorization results are compelling (with comparison with previous methods), while clearly showing the failure cases where the model should be improved. Ablation studies are done to show the importance of different components (e.g. the conditioning network). The paper is clearly written.

A few remarks:
- arctan soft-clamping seems very similar to the scalar times tanh soft-clamping of Real NVP (Dinh et al., 2016). Why was arctan adopted?
- the choice of the car image (the biggest one in Figure 10) for the colorization transfer is questionable. I'm not able to tell from this figure if there was any segmentation happening in the model. The pose of the cars are similar, the car in the back is mostly black. The colorization transfer result gives me the impression that the segmentation is not done properly, e.g. the red color from the red car image seems to spill outside of the confine of the car in the colorization transfer.

**Experience Assessment:**

I have published in this field for several years.

**Review Assessment: Checking Correctness Of Derivations And Theory:**

N/A

**Review Assessment: Checking Correctness Of Experiments:**

I assessed the sensibility of the experiments.

**Review Assessment: Thoroughness In Paper Reading:**

N/A

---

> ### Author Response · Authors · 2019-11-14
> **Review Response**
>
> Thank you for taking the time to review our paper, and for the insightful remarks.
> We address your points below. We have also updated the PDF, and highlighted the changes in red.
>
> Concerning the soft-clamping in RealNVP:
> Thank you for pointing this out! In fact, we had overlooked this until now (it is only briefly mentioned in the RealNVP paper, and details have to be extracted from the code).
> The main difference to our work is that alpha is a scalar hyperparameter in our cINN, but a learned vector in RealNVP. This should not limit the expressive power of our architecture, because the adaptive scaling is simply absorbed into the previous layer. The choice between arctan and tanh was not deliberate, we expect them to work equally well. We have revised the text accordingly.
>
> Concerning the car image for color transfer:
> The car image was chosen because it illustrates color transfer very intuitively.
> We have included an additional example in the appendix, which demonstrates that the cINN segments objects correctly when transferring colors.

---

### Decision · Program_Chairs · 2019-12-19

**Decision:**

Reject

**Comment:**

The paper presents an extension of flow-based invertible generative models to a conditional setting. The key idea is fairly simple modification of the original architecture, but authors also propose techniques for down-sampling with Haar wavelets. The experimental results on class-conditional MNIST generation and colorization are promising. However, in terms of weakness, the technical novelty seems somewhat limited although it's a reasonable extension. In addition, the experimental results lack evaluation on general conditional image generation tasks with more widely used benchmarks (e.g., class-conditional generation setting for real images, such as CIFAR and ImageNet; attribute-conditional or image-to-image translation settings; etc.). In other words, colorization seems like a niche task. The baselines compared are not the strongest models. For example, the diversity of
cGANs can be significantly improved by simple plug-in modifications (e.g., DSGAN) to any existing GAN architectures, and those methods were demonstrated on broader benchmarks. So I view the experimental validation somewhat limited in scope and significance. While this work presents a reasonable extension of conditional invertible generative models with promising results, I believe that more work needs to be done to be publishable at a top-tier conference.

Diversity-Sensitive Conditional Generative Adversarial Networks
https://arxiv.org/abs/1901.09024

Mode Seeking Generative Adversarial Networks for Diverse Image Synthesis
https://arxiv.org/abs/1903.05628
* exactly the same idea as DSGAN above.